# SENP1 prevents steatohepatitis by suppressing RIPK1-driven apoptosis and inflammation

Lingjie Yan[1,2,9], Tao Zhang[3,9], Kai Wang [4,5,9], Zezhao Chen[1,2], Yuanxin Yang[1,2], Bing Shan[1], Qi Sun[1], Mengmeng Zhang[1], Yichi Zhang[6], Yedan Zhong[1], Nan Liu[1,7], Jinyang Gu [8,6] ✉ & Daichao Xu [1,7] ✉

Activation of RIPK1-driven cell death and inflammation play important roles in the progression of nonalcoholic steatohepatitis (NASH). However, the mechanism underlying RIPK1 activation in NASH remains unclear. Here we identified SENP1, a SUMO-specific protease, as a key endogenous inhibitor of RIPK1. SENP1 is progressively reduced in proportion to NASH severity in patients. Hepatocyte-specific SENP1-knockout mice develop spontaneous NASH-related phenotypes in a RIPK1 kinase-dependent manner. We demonstrate that SENP1 deficiency sensitizes cells to RIPK1 kinase-dependent apoptosis by promoting RIPK1 activation following TNFα stimulation. Mechanistically, SENP1 deSUMOylates RIPK1 in TNF-R1 signaling complex (TNF-RSC), keeping RIPK1 in check. Loss of SENP1 leads to SUMOylation of RIPK1, which re-orchestrates TNF-RSC and modulates the ubiquitination patterns and activity of RIPK1. Notably, genetic inhibition of RIPK1 effectively reverses disease progression in hepatocyte-specific SENP1-knockout male mice with high-fat-diet-induced nonalcoholic fatty liver. We propose that deSUMOylation of RIPK1 by SENP1 provides a pathophysiologically relevant cell death-restricting checkpoint that modulates RIPK1 activation in the pathogenesis of nonalcoholic steatohepatitis.

Nonalcoholic steatohepatitis (NASH) is a combination of hepatic steatosis, severe inflammation, and liver injury, and it often follows simple hepatic steatosis in nonalcoholic fatty liver disease (NAFLD)[1]. Hepatocellular death, characterized by swollen hepatocytes on liver biopsy, and hepatic inflammation are cardinal features of NASH[2,3]. Improving liver damage induced by hepatocellular death and preventing inflammation are major goals of NASH therapy[3]. Currently, there are no approved effective pharmacological therapies for NASH, and efforts to control the complications arising from the condition are far from satisfactory[4]. Considering that the pathogenesis of NASH is a complex process involving chronic inflammation and liver damage, the ideal drug target(s) should be signaling factor(s) that mediate these major pathophysiological pathways.

RIPK1 is a serine/threonine kinase that functions as a master upstream regulator of cell death and inflammation in cells stimulated by TNFα[5,6]. TNFα is an important pro-inflammatory cytokine that can

[1]Interdisciplinary Research Center on Biology and Chemistry, Shanghai Institute of Organic Chemistry, Chinese Academy of Sciences, Shanghai 201210, China. [2]University of Chinese Academy of Sciences, Beijing 100049, China. [3]Department of Pathology, Beth Israel Deaconess Medical Center, Harvard Medical School, Boston, MA 02215, USA. [4]Department of Hepatobiliary and Pancreatic Surgery, Key Laboratory of Integrated Oncology and Intelligent Medicine of Zhejiang Province, Hangzhou First People's Hospital Affiliated Zhejiang University School of Medicine, Hangzhou 310006, China. [5]Institute of Organ Transplantation, Zhejiang University, Hangzhou 310003, China. [6]Department of Transplantation, Xinhua Hospital Affiliated to Shanghai Jiao Tong University School of Medicine, Shanghai 200092, China. [7]Shanghai Key Laboratory of Aging Studies, Shanghai 201210, China. [8]Center for Liver Transplantation, Union Hospital, Tongji Medical College, Huazhong University of Science and Technology, Wuhan 430022, China. [9]These authors contributed equally: Lingjie Yan, Tao Zhang, Kai Wang. ✉e-mail: gjynyd@126.com; xudaichao@sioc.ac.cn

induce hepatocellular death, and critically involved in the pathogenesis and disease progression of NASH[7–11]. Activation of RIPK1 downstream of TNFR1 signaling plays an important role in promoting programmed cell death via either necroptotic or apoptotic pathway[5,6]. In TNFα-stimulated cells, RIPK1 is rapidly recruited into the TNF-R1 signaling complex (TNF-RSC) where RIPK1 is extensively modulated by an intricate set of ubiquitination which collectively decides whether RIPK1 is to be activated[12–15]. Activation of RIPK1 kinase promotes much of the deleterious effects activated by TNFα in human diseases[5,6,16]. RIPK1 activation has been found in both mouse models of NASH and human patients with NASH[17,18]. Inhibition of RIPK1 kinase can reduce hepatic inflammation and liver damage in experimental NASH[17,18], highlighting the importance of understanding the molecular mechanisms that control RIPK1 activation in NASH.

Conjugation of small ubiquitin-like modifier (SUMO) to a large number of substrates has been suggested to regulate numerous signaling pathways through alterations of its targeting protein functions[19]. The SUMO family is composed of SUMO1-3, in which SUMO2 and SUMO3 share 97% sequence identity and to form poly-SUMO chains[20,21]. SUMOylation has been proven to play key roles in the pathogenesis of NASH[22–25], suggesting that controlling SUMOylation of key molecular targets may provide a promising avenue for NASH pharmacotherapies. SUMOylation can be readily reversed by a family of SUMO-specific proteases (SENPs), among which, SENP1 is largely responsible for the deconjugation of both SUMO1 and SUMO2/3 modifications in a large number of target proteins[26]. SENP1 is integrally involved in a variety of physiological processes, including hypoxia signaling[27], angiogenesis[28], and erythropoiesis[29] via deSUMOylation of distinct targets. SENP1 has been shown to limit inflammatory response in type 1 diabetes, and fibrotic responses in liver fibrosis[30,31]. $Senp1^{-/-}$ mice die around E14.5 due to fetal liver cell death, anemia and hypoplasia[27,29]. Although SENP1 functions in many different systems, its pathophysiological role has not been well defined.

In this study, we show that SENP1 serves an essential function in TNFR1 signaling to control the activation of RIPK1. SENP1 is markedly downregulated in the livers of individuals with NAFLD or NASH and that SENP1 absence from mouse hepatocytes hastens the development of spontaneous NASH-related phenotypes, including liver inflammation, lipid accumulation, liver damage, and fibrosis, in a RIPK1 kinase-dependent manner. SENP1 deficiency sensitizes cells to RIPK1 kinase-dependent apoptosis following TNFα stimulation, and genetic inactivation of RIPK1 by kinase-dead D138N knockin mutation rescues the cell death phenotype of $Senp1^{-/-}$ fetal liver. We demonstrate that SENP1 directly binds to and deSUMOylates RIPK1 in vitro and in cells. We show that SUMOylation of RIPK1 at Lys550 in SENP1-deficient cells reorchestrates TNF-RSC by reducing the recruitment of A20 whereas promoting that of LUBAC complex to TNF-RSC and modulates the ubiquitination patterns and activity of RIPK1. We also show that the SUMO E3 ligase PIAS1 is involved in performing SUMOylation on RIPK1 in SENP1-deficient cells. Moreover, liver-specific SENP1 knockout exaggerates hepatic inflammation and liver damage in high-fat-diet (HFD)-induced nonalcoholic fatty liver, which is suppressed by RIPK1-D138N mutation. Collectively, our data identify SENP1-mediated RIPK1 deSUMOylation as a pathophysiologically relevant cell death-restricting checkpoint in TNF-RSC and reveal a role for SENP1 in NASH pathogenesis.

## Results

### Reduced SENP1 expression in fatty liver is associated with the severity of NAFLD

Given an intimate association of SENP1 with inflammation and fibrosis, as well as the apparent functional role of SUMOylation in NASH, we hypothesized that SENP1 has a pivotal role in NASH pathogenesis as well. To evaluate whether SENP1 was involved in the progression of NAFLD, we first examined its expression in the livers of human subjects without NAFLD, with NAFLD and NASH. We found that the protein

levels of SENP1 were substantially decreased in the livers of subjects with simple steatosis or NASH, as compared to that in the nonsteatotic controls (Fig. 1a). Furthermore, the NASH group had considerably lower SENP1 expression than the simple steatosis group (Fig. 1a). Moreover, hepatic SENP1 protein levels in human subjects were negatively correlated with NASH activity score (NAS), serum levels of alanine transaminase (ALT) and aspartate aminotransferase (AST) (Fig. 1b, c). In addition, expression of SENP1 in mouse liver was also markedly decreased through feeding with a high-fat diet (HFD) (Supplementary Fig. 1a).

### Hepatocyte-specific SENP1-knockout mice develop spontaneous liver damage in RIPK1 kinase-dependent manner

We found that SENP1 is specifically reduced in hepatocytes during NAFLD (Supplementary Fig. 1b). we then generated hepatocyte-specific SENP1-knockout mice by crossing $Senp1^{f/f}$ mice with Albumin-Cre (Alb-Cre) Tg mice to elucidate the function of SENP1 in NAFLD. The efficient deletion of SENP1 in hepatocytes was confirmed by immunoblot analysis of whole liver and isolated primary hepatocytes (Supplementary Fig. 1c, d). We found no mortality in the $Senp1^{f/f}$;Alb-Cre mice up to eight months, at which time liver damage was observed under unstressed conditions (Fig. 1d). Hepatocytic damage can be directly measured by ALT and AST levels in serum[32]. We found that ALT and AST levels in the serum of $Senp1^{f/f}$;Alb-Cre mice were elevated compared with that of $Senp1^{f/f}$ mice (Fig. 1e). Moreover, we detected substantially increased apoptotic cells in $Senp1^{f/f}$;Alb-Cre mice at eight months of age as determined by terminal deoxynucleotidyl transferase-mediated deoxyuridine triphosphate nick end labeling (TUNEL) staining (Fig. 1f).

Knockdown of SENP1 has been found in a focused genome-wide siRNA screen to sensitize to RIPK1-dependent cell death in mouse fibroblast L929 cells[33] (Z-score = −3.88) (Supplementary Fig. 1e), suggesting a potential role of SENP1 in suppressing RIPK1 activation. Given the role of RIPK1 kinase in driving the progression of NASH in both mouse models and human patients, we next examined whether RIPK1 activation can be seen in $Senp1^{f/f}$;Alb-Cre mice using p-S166 RIPK1 (activation biomarker of RIPK1[34–36]) immunostaining and immunoblotting. We detected RIPK1 activation as determined by substantial amount of p-S166 RIPK1 positive cells in liver sections of $Senp1^{f/f}$;Alb-Cre mice and increased p-S166 RIPK1 in primary hepatocytes isolated from 8 months old $Senp1^{f/f}$;Alb-Cre mice (Fig. 1g and Supplementary Fig. 1f). To address the function of RIPK1 activation in contributing to liver damage of $Senp1^{f/f}$;Alb-Cre mice, we next generated $Senp1^{f/f}$;Alb-Cre;$Ripk1^{D138N/D138N}$ mice by crossing $Senp1^{f/f}$;Alb-Cre mice with RIPK1 kinase-dead knock-in $Ripk1^{D138N/D138N}$ mice[37]. We found that inactivating RIPK1 substantially prevented liver damage and hepatocellular death in $Senp1^{f/f}$;Alb-Cre;$Ripk1^{D138N/D138N}$ mice as compared with $Senp1^{f/f}$;Alb-Cre mice (Fig. 1d–f). The death of hepatocytes is mainly apoptosis as determined by increased levels of cleaved caspase 3 (CC3) in SENP1-deficient hepatocytes, which is blocked in $Senp1^{f/f}$;Alb-Cre;$Ripk1^{D138N/D138N}$ hepatocytes (Supplementary Fig. 1f). However, we did not detect apparent signal of p-T231/S232 RIPK3, the hallmark of necroptosis[5,6], by immunoblotting in SENP1-deficient hepatocytes (Supplementary Fig. 1f). Thus, RIPK1 activation promotes hepatocellular apoptosis in $Senp1^{f/f}$;Alb-Cre mice.

### Deletion of SENP1 in hepatocytes causes RIPK1-driven inflammation, steatosis, and fibrosis

We next characterized the early effect of hepatic SENP1 deletion. We found increased inflammatory CD45+ cell infiltration in $Senp1^{f/f}$;Alb-Cre mice at early stage (3 months of age) (Supplementary Fig. 1g). At 8 months of age, $Senp1^{f/f}$;Alb-Cre mice displayed significant infiltration of CD45+ cells compared with control mice, which was suppressed by RIPK1-D138N mutation (Fig. 2a). Hepatic mRNA expression of inflammatory genes, including Il1a/b, Tnf, and Cxcl1,

were increased in 3-month-old *Senp1^{f/f};Alb-Cre* mice compared with WT littermates (Supplementary Fig. 1h). We next conducted total liver RNA-seq to verify this result. The gene transcriptional patterns from the livers of 3-month-old *Senp1^{f/f};Alb-Cre* mice showed dramatic changes compared to that of *Senp1^{f/f}* mice (Supplementary Fig. 1i). Gene ontology (GO) analysis showed that hepatic SENP1 deficiency led to significant increase in the expression levels of those genes involved in mediating inflammatory response, such as TNFR1 signaling and NF-κB pathway (genes including *Sharpin, Nfkb1, Tnfrsf1a,* and *Tnfrsf9*), and apoptotic process (Fig. 2b and Supplementary Fig. 1j). In addition, multiple homeostatic and metabolic pathways were downregulated in *Senp1^{f/f};Alb-Cre* mice

(Supplementary Fig. 1k). Strikingly, the up-regulation of these pro-inflammatory genes, including *Tnf, Il1a/b, Cxcl1, Ccl2,* and *Ccl5* were restored to that of *Senp1^{f/f}* control levels in *Senp1^{f/f}; Alb-Cre;Ripk1^{D138N/D138N}* livers and primary hepatocytes (Fig. 2c and Supplementary Fig. 1l).

Hepatic inflammation has been shown to promote steatosis in NAFLD. We found that *Senp1^{f/f};Alb-Cre* mice displayed steatosis, as determined by oil red O (ORO) staining, as early as at 3-month of age (Supplementary Fig. 1m), and remained elevated at 8 months old (Fig. 2d). The deposition of lipid in *Senp1^{f/f};Alb-Cre* livers and primary hepatocytes was further confirmed by measuring hepatic concentrations of triglyceride (TG) and total cholesterol (TC) (Fig. 2e and

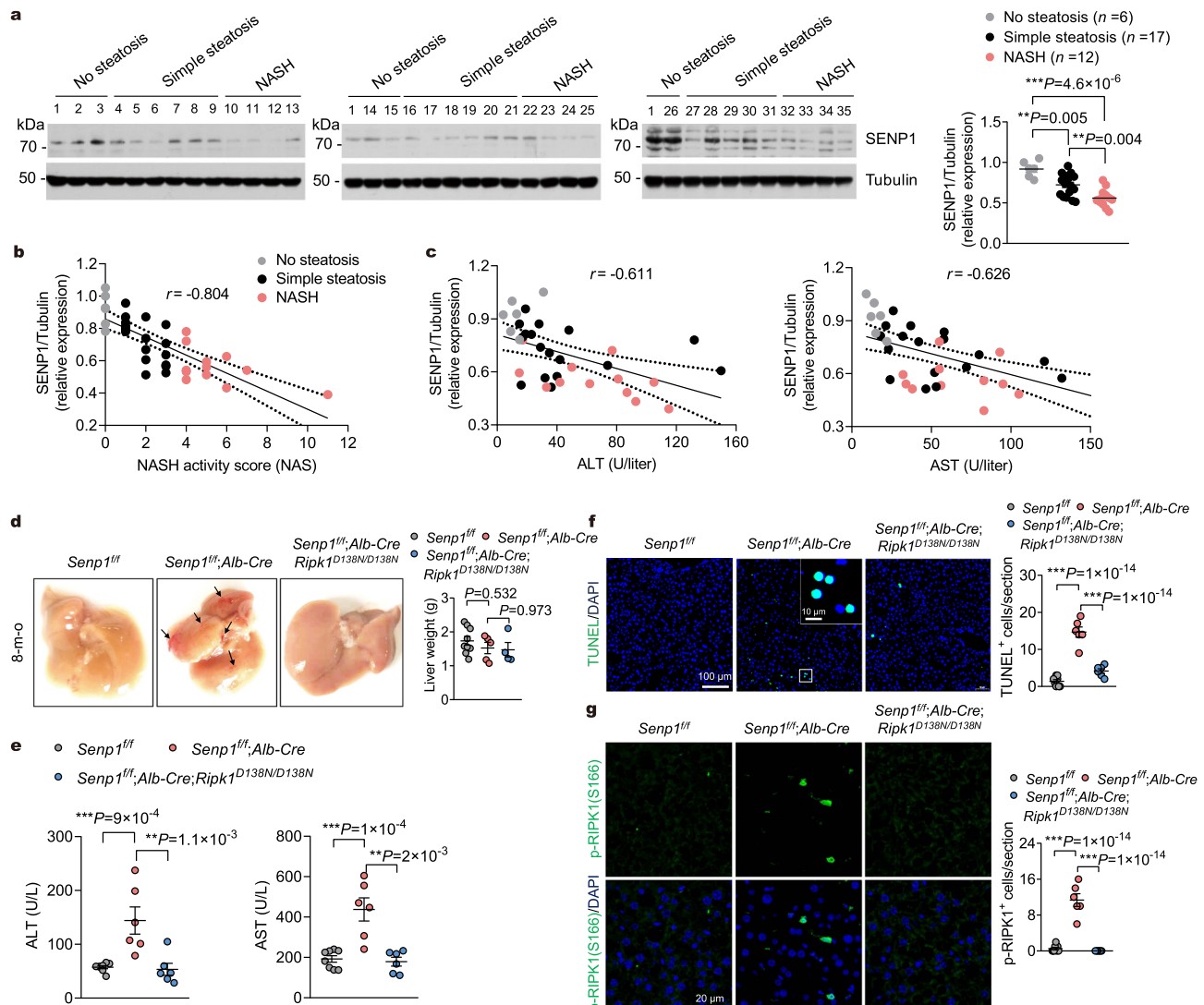

**Fig. 1 | SENP1 is downregulated in fatty liver, deletion of SENP1 in hepatocytes causes RIPK1-dependent liver damage. a** Western blots and quantification of SENP1 expression in the livers of individuals with non-steatosis (*n* = 6), simple steatosis (*n* = 17) or NASH (*n* = 12). Protein expression was normalized to Tubulin levels and shown as relative values. Pearson comparison analyses of the correlation between SENP1 protein levels and NASH activity score (NAS) (**b**), serum alanine transaminase (ALT), and aspartate aminotransferase (AST) concentrations (**c**). Two-tailed Spearman's rank correlation coefficient analysis ($P = 1 \times 10^{-14}$) (**b, c**). **d** Gross appearance of livers from 8-month-old mice of indicated genotypes. Arrows indicate liver damage and inflammation. Representative images out of *n* = 9 (*Senp1^{f/f}*), *n* = 5 (*Senp1^{f/f};Alb-Cre*), and *n* = 4 (*Senp1^{f/f};Alb-Cre;Ripk1^{D138N/D138N}*) mice are represented. Liver weight of indicated genotypes is shown on right. **e** Serum levels of ALT and AST from 8-month-old mice of indicated genotypes (*n* = 8 for *Senp1^{f/f}*, *n* = 6 for

*Senp1^{f/f};Alb-Cre,* and *Senp1^{f/f};Alb-Cre;Ripk1^{D138N/D138N}*). **f** Immunofluorescence images of p-S166 RIPK1 of liver sections from 8-month-old mice with indicated genotypes. Representative images out of *n* = 8 (*Senp1^{f/f}*), *n* = 6 (*Senp1^{f/f};Alb-Cre* and *Senp1^{f/f};Alb-Cre;Ripk1^{D138N/D138N}*) mice are represented. Graph depicting numbers of p-S166 RIPK1^+ cells on liver sections of indicated genotypes. **g** Terminal deoxynucleotidyl transferase-mediated deoxyuridine triphosphate nick end labeling (TUNEL) assay was performed on liver sections from 8-month-old mice with indicated genotypes. Representative images out of *n* = 8 (*Senp1^{f/f}*), *n* = 6 (*Senp1^{f/f};Alb-Cre* and *Senp1^{f/f};Alb-Cre;Ripk1^{D138N/D138N}*) mice are represented. Graph depicting numbers of TUNEL^+ cells on liver sections of indicated genotypes. Data are expressed as mean ± s.e.m. (**a, d, e, g**). One-way ANOVA, post hoc Dunnett's test (**a, d, e, f, g**). Source data are provided as a Source Data file.

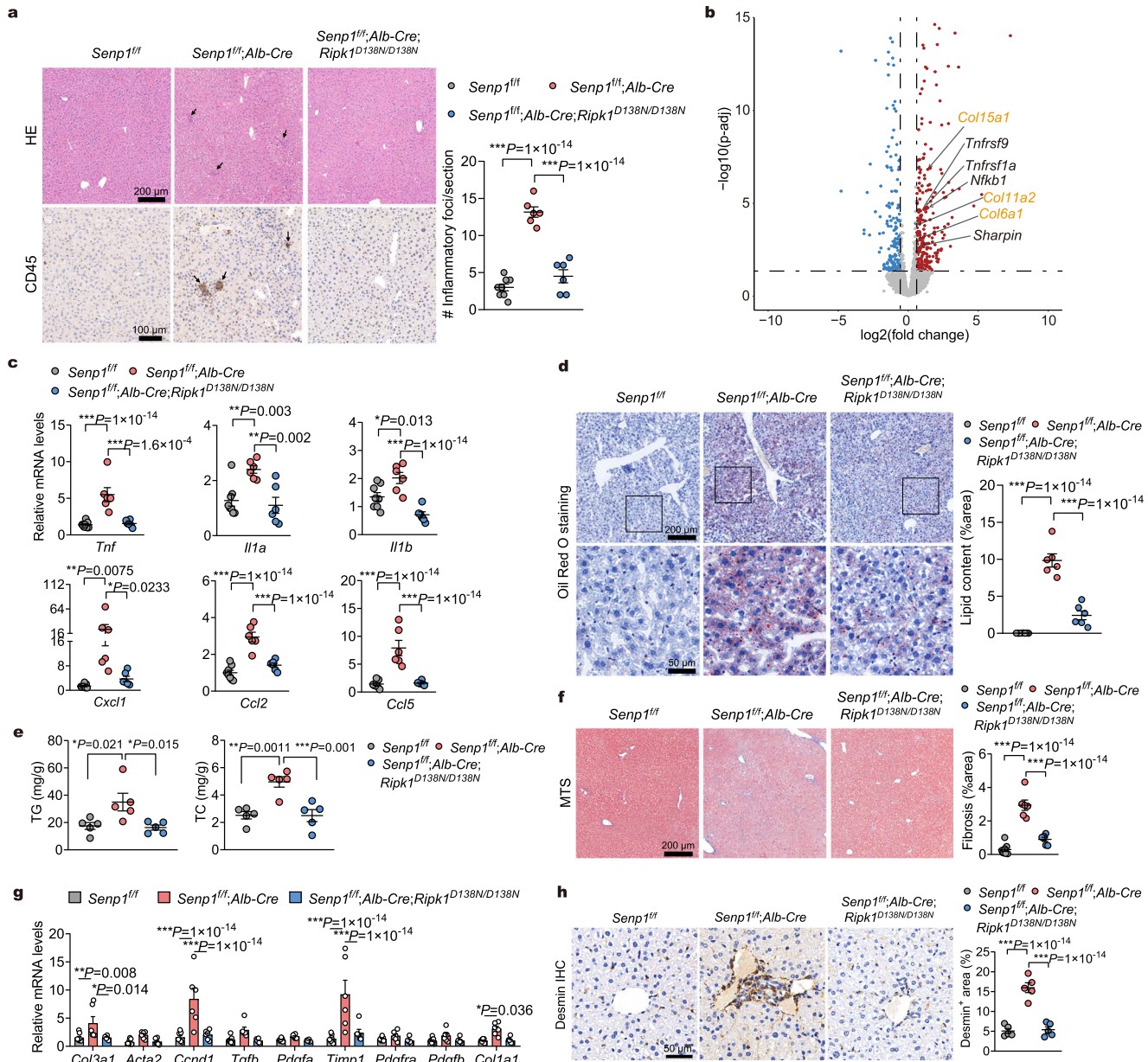

**Fig. 2 | Deletion of SENP1 in hepatocytes causes RIPK1-driven inflammation, steatosis, and fibrosis. a** Hematoxylin and eosin (HE) staining and immunohistochemistry (IHC) of CD45 of liver sections from 8-month-old mice of indicated genotypes. Representative images out of $n = 8$ (*Senp1*^f/f^), $n = 6$ (*Senp1*^f/f^;*Alb-Cre* and *Senp1*^f/f^;*Alb-Cre*;*Ripk1*^D138N/D138N^) mice are represented. Arrows indicate infiltration of macrophages. Graph depicting numbers of CD45^+^ foci on liver sections of indicated genotypes. **b** Volcano plots showing fold-change and *P*-value for the comparison of 3-month-old *Senp1*^f/f^;*Alb-Cre* livers versus *Senp1*^f/f^ livers ($n = 3$). Differentially expressed genes (fold change >1.5, *P* value < 0.05) are depicted in red (upregulated) and blue (downregulated). Representative genes associated with inflammatory response (TNFα signaling, black) and fibrosis (collagen, yellow) are highlighted. **c**, Quantitative reverse transcription-polymerase chain reaction (RT-PCR) analysis of the mRNA expression of cytokines and chemokines from livers of 8-month-old mice of indicated genotypes ($n = 8$ for *Senp1*^f/f^, $n = 6$ for *Senp1*^f/f^;*Alb-Cre* and *Senp1*^f/f^;*Alb-Cre*;*Ripk1*^D138N/D138N^). **d** Oil Red O staining of lipids in liver sections from 8-month-old mice of indicated genotypes. Representative images out

of $n = 8$ (*Senp1*^f/f^), $n = 6$ (*Senp1*^f/f^;*Alb-Cre* and *Senp1*^f/f^;*Alb-Cre*;*Ripk1*^D138N/D138N^) mice are represented. Graph depicting percentage of lipid content on liver sections of indicated genotypes. **e** Triglyceride (TG) and total cholesterol (TC) concentrations in the livers of mice from the indicated groups ($n = 5$). **f** Representative images of Masson's trichrome stained (MTS) liver sections from 8-month-old mice of indicated genotypes. Representative images out of $n = 8$ (*Senp1*^f/f^), $n = 6$ (*Senp1*^f/f^;*Alb-Cre* and *Senp1*^f/f^;*Alb-Cre*;*Ripk1*^D138N/D138N^) mice are represented. Graph depicting percentage of fibrosis area on liver sections of indicated genotypes. **g** Quantitative RT-PCR analysis of the mRNA expression of fibrogenic parameters from livers of 8-month-old mice of indicated genotypes ($n = 8$ for *Senp1*^f/f^, $n = 6$ for *Senp1*^f/f^;*Alb-Cre* and *Senp1*^f/f^;*Alb-Cre*;*Ripk1*^D138N/D138N^). **h** Immunohistochemistry of hepatic stellate cells (HSC) marker Desmin in the livers of indicated genotypes. Representative images out of $n = 5$ mice for each genotype are represented. Data are expressed as mean ± s.e.m. (**a**, **c**–**h**). One-way ANOVA, post hoc Dunnett's test (**a**, **c**–**h**). Source data are provided as a Source Data file.

Supplementary Fig. 1n). Interestingly, inhibition of RIPK1 in *Senp1*^f/f^;*Alb-Cre*;*Ripk1*^D138N/D138N^ mice strongly suppressed steatosis (Fig. 2d, e and Supplementary Fig. 1n).

Hepatocytic damage and liver inflammation are known to activate hepatic stellate cells (HSC), which then release extracellular matrix proteins, including collagen, to promote the formation of fibrotic

scars[38–40]. We noted that in the livers of *Senp1^f/f;Alb-Cre* mice at 3 months of age, the mRNA levels of fibrillary collagen, including *Col11a2, Col6a1,* and *Col15a1,* were increased compared with *Senp1^f/f* mice (Fig. 2b). The fibrillary collagen deposition in the livers of *Senp1^f/f;Alb-Cre* mice was confirmed by Masson's trichrome staining (MTS) (Fig. 2f and Supplementary Fig. 1o). Because inhibition of RIPK1 in *Senp1^f/f;Alb-Cre;Ripk1^D138N/D138N* mice suppressed hepatocytic damage and liver inflammation, we found that the deposition of collagen in *Senp1^f/f;Alb-Cre;Ripk1^D138N/D138N* mice were also reduced (Fig. 2f). In addition, mRNA expression of fibrogenic parameters, which were upregulated in the livers of *Senp1^f/f;Alb-Cre* mice when compared with that of controls, were reduced in the livers of *Senp1^f/f;Alb-Cre;Ripk1^D138N/D138N* mice (Fig. 2g). We next analyzed the expression of Desmin, the marker of HSC, in the livers of mice. Consistent with the upregulation of the HSC activation genes *Col1a1,* and *Col3a1,* the numbers of HSC were increased in *Senp1^f/f;Alb-Cre* mice compared with that of *Senp1^f/f* mice, which was also reduced in *Senp1^f/f;Alb-Cre;Ripk1^D138N/D138N* mice (Fig. 2h). These results indicate that HSC activation induced by liver injury and inflammation in *Senp1^f/f;Alb-Cre* mice were regulated by RIPK1 kinase activity. Thus, these above results demonstrated the essential role of RIPK1 kinase activity in mediating liver inflammation and damage, including steatosis and fibrosis, under a hepatocyte-specific SENP1-deficiency condition.

We next determined whether the effect of SENP1 in preventing steatohepatitis and liver damage depends on its deSUMOylation activity. To this end, we used liver-specific adeno-associated virus (AAV) to reconstitute WT and catalytically inactive C599S mutant of murine SENP1 in the lives of 2 months old *Senp1^f/f;Alb-Cre* mice, respectively. After AAV infection for 2 months (Supplementary Fig. 2a), we detected RIPK1 activation in livers of *Senp1^f/f;Alb-Cre;Senp1-C599S* (SENP1-C599S) mice, but not that of SENP1-WT mice (Supplementary Fig. 2b). Consistently, SENP1-C599S but not SENP1-WT livers showed CC3^+ cells (Supplementary Fig. 2c). SENP1-C599S livers also showed elevated inflammatory cytokine production than that of SENP1-WT livers (Supplementary Fig. 2d). SENP1-C599S but not SENP1-WT livers display apparent steatosis (Supplementary Fig. 2e). Taken together, these results suggest that the deSUMOylation activity of SENP1 is important in suppressing RIPK1 activation, and subsequent apoptosis, inflammation, and steatosis.

## SENP1 deficiency promotes RIPK1 activation and cell death
To further support the role of SENP1 in suppressing RIPK1 activation, we used short interfering RNAs (siRNAs) to knock down 14 rate-limiting proteins that are involved in SUMOylation process, including six SUMO-specific proteases, five SUMO E3 ligases, and SUMO1/2/3, in mouse embryonic fibroblast cells (MEFs) stimulated with TNFα/TAK1 inhibitor 5Z-7-Oxozeaenol (5z7) to induce RIPK1-dependent apoptosis (RDA)[41,42]. Consistent with previous study[33], SENP1 knockdown (KD) greatly sensitized cells to RDA (Fig. 3a and Supplementary Fig. 3a), which was also confirmed in cells expressing SENP1-specific short hairpin RNA (shRNA) (Supplementary Fig. 3b). The death of SENP1-KD MEFs induced by TNFα/5z7 was fully protected by RIPK1 kinase inhibitor necrostatin-1s (Nec-1s)[35] (Supplementary Fig. 3c). SENP1 KD enhanced RIPK1 activation, as shown by increased activation biomarker p-S166 RIPK1, and apoptosis as shown by increased levels of the cleaved-caspase-3 (CC3) (Supplementary Fig. 3d). Thus, SENP1 KD promotes RIPK1 activation and RDA in response to pro-RDA stimuli.

The activation of RIPK1 in a kinase-independent manner can promote cell survival mediated by NF-κB pathway; alternatively, the activation of RIPK1 kinase drives cell death and inflammation under certain conditions in response to TNFα stimulation[6]. We found that SENP1 KD has minor effect on NF-κB and MAPK pathway activation in TNFα-treated MEFs (Supplementary Fig. 3e). Interestingly, SENP1 KD sensitized to cell death induced by TNFα alone, a condition that cannot induce the death of wild-type (WT) MEFs (Fig. 3b and Supplementary

Fig. 3f, g). SENP1 KD enhanced RIPK1 activation and apoptosis in response to TNFα stimulation, as marked by p-S166 RIPK1 and CC3, respectively (Fig. 3c). In addition, SENP1 deficiency in primary hepatocytes also enhanced RIPK1 activation and sensitized to apoptosis induced by TNFα alone (Fig. 3d, e). Thus, SENP1 deficiency unusually sensitized cells to RDA when stimulated with TNFα alone.

Cycloheximide (CHX) blocks the translation downstream of NF-κB pathway activated by TNFα and promotes RIPK1-independent apoptosis (RIA) of WT MEFs when treated with TNFα, which cannot be blocked by RIPK1 inhibitor Nec-1s[43,44] (Supplementary Fig. 4a). In contrast, SENP1-KD MEFs showed greater levels of cell death in response to TNFα/CHX, which was blunted by treatment with Nec-1s (Supplementary Fig. 4a). Furthermore, TNFα/CHX-treated SENP1-KD MEFs, but not WT MEFs, induced p-S166 RIPK1 (Supplementary Fig. 4b). Thus, SENP1 KD not only sensitizes cells to RDA induced by TNFα alone, but also converts RIA induced by TNFα/CHX into RDA.

When caspases are inhibited by the pan-caspase inhibitor zVAD.fmk (zVAD), the kinase activity of RIPK1 is activated to interact with RIPK3 to induce the formation of a RIPK1/RIPK3 complex (known as necrosome) and the activated RIPK3 in turn mediates the phosphorylation of MLKL to promote the execution of necroptosis[45]. We found that SENP1 KD greatly sensitized cells to TNFα/CHX/zVAD-induced necroptosis (Supplementary Fig. 4c). The increased sensitivity of SENP1-KD MEFs to necroptosis was also marked by the increased levels of the necroptotic biomarkers p-S166 RIPK1, p-T231/S232 RIPK3, and p-S345 MLKL relative to that of WT MEFs (Supplementary Fig. 4d). Consistent with increased activation of necroptosis, we found that SENP1-KD MEFs showed more necrosome formation than that of WT MEFs (Supplementary Fig. 4e). Thus, SENP1 KD promotes RIPK1 activation in TNFR1 signaling and SENP1-KD cells are hypersensitized to RIPK1-dependent forms of cell death mediated by TNFα.

## SENP1 suppresses RIPK1 activation during embryonic fetal liver development
Since SENP1 deficiency in mice leads to embryonic lethality due to massive fetal liver apoptosis and anemia after post-coital day E14.5[27], we considered the possibility that RIPK1 activation might contribute to the cell death phenotype of SENP1-deficient mice. We detected substantial increases in activated p-S166 RIPK1^+ cells throughout *Senp1^−/−* fetal liver (Fig. 3f and Supplementary Fig. 5a). We then generated *Senp1^−/−;Ripk1^D138N/D138N* mice to study the contribution of abnormal activation of the kinase activity of RIPK1 to the liver apoptosis phenotype of *Senp1^−/−* mice. We performed histological analysis and TUNEL assays to sections from E14.5 WT and *Senp1^−/−* embryos. *Senp1^−/−* embryos showed severe liver degeneration with a large number of TUNEL^+ cells (Fig. 3g, h). *Senp1^−/−* fetal livers exhibited activated caspase-3 as assessed by immunohistochemistry of CC3 throughout the liver (Fig. 3i). Interestingly, these hallmarks of apoptosis were largely blocked by RIPK1-D138N mutation (Fig. 3g–i).

We also observed significant increases in the levels of pro-inflammatory cytokines and chemokines in *Senp1^−/−* fetal livers, including *Il1a/b, Tnf, Cxcl1,* and *Ccl2,* which were reduced in *Senp1^−/−;Ripk1^D138N/D138N* embryos (Fig. 3j). We next analyzed the expression profile of E14.5 fetal livers by RNA sequencing (RNA-seq). Compared to WT fetal livers, *Senp1^−/−* fetal livers showed markedly elevated expression of genes involved in apoptotic process and immune response, which is in line with the increased levels of cell death and pro-inflammatory cytokines and chemokines in *Senp1^−/−* fetal livers (Supplementary Fig. 5b, c). *Ripk1* mRNA levels can be upregulated during RIPK1-dependent cell death in certain cells[46]. Interestingly, *Senp1^−/−* fetal livers also showed elevated expression of mRNA levels of *Ripk1,* which was confirmed by quantitative RT-PCR (Fig. 3k, l). The elevated expression of *Ripk1* mRNA levels was reduced in *Senp1^−/−;Ripk1^D138N/D138N* fetal livers (Fig. 3l). *Senp1^−/−* embryos have poor blood vessel development and anemia[29]; consistently, *Senp1^−/−* fetal

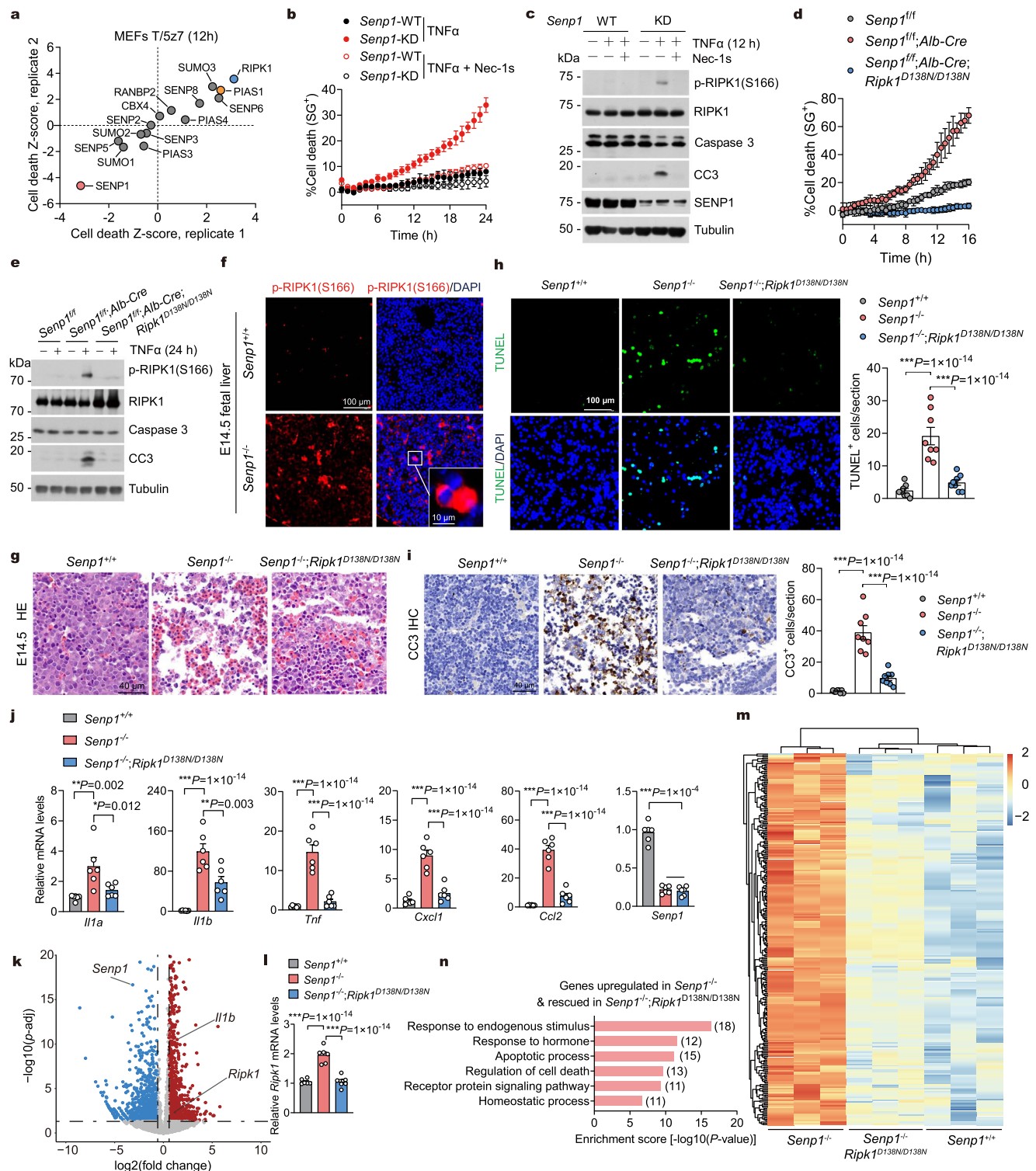

livers showed downregulation of multiple blood vessel development pathways (Supplementary Fig. 5d). Interestingly, the gene expression abnormalities of 278 genes (~30% genes upregulated in $Senp1^{-/-}$ relative to WT), including these involved in apoptotic process and immune response, were restored to WT levels by genetic inhibition of RIPK1 in $Senp1^{-/-};Ripk1^{D138N/D138N}$ fetal livers (Fig. 3m, n and Supplementary Fig. 5e). Thus, SENP1 suppresses RIPK1 activation and RDA during embryonic fetal liver development.

However, we were unable to obtain viable $Senp1^{-/-};Ripk1^{D138N/D138N}$ mice despite the prevention of cell death in fetal livers of these mice (Supplementary Fig. 5f). Histological analysis of $Senp1^{-/-};Ripk1^{D138N/D138N}$

fetal lives showed a disproportionate abundance of immature erythroblasts as seen in $Senp1^{-/-}$ fetal livers, compared with that of WT fetal livers (Fig. 3g), suggesting that erythropoiesis defects leading to anemia in $Senp1^{-/-}$ embryos was not rescued by RIPK1-D138N mutation. Similarly, RIPK1-D138N mutation does not alter the downregulation of genes that are involved in blood vessel development in $Senp1^{-/-}$ embryos (Supplementary Fig. 5g, h). In addition, $Senp1^{-/-}$ embryos show reduced expression of erythropoietin (EPO)[27], which is important for hematopoiesis. We found that RIPK1-D138N mutation could not rescue the deficient EPO production in $Senp1^{-/-};Ripk1^{D138N/D138N}$ embryos (Supplementary Fig. 5i). Defective rescue of the expression of EPO and its

**Fig. 3 | SENP1 suppresses RIPK1-dependent apoptosis during embryonic fetal liver development. a** MEFs were transfected with siRNA pools targeting 14 rate-limiting enzymes/proteins that are involved in SUMOylation process for 48 h followed by treatment with TNFα (T, 1 ng/ml) and (5Z)−7-oxozeaeno (5z7, 100 nM) and cultured for an additional 12 h to induce cell death. Cell death was then measured and Z-score was calculated as described. The screen hits were selected based on the median Z-score of the duplicate plates with cut-offs set at Z-score >2 or <−2. Blue, RIPK1 (positive control). Red, SENP1. Yellow, PIAS1. MEFs (**b**, **c**) and primary hepatocytes derived from 2-m-o mice (**d**, **e**) were treated with 10 ng/ml TNFα in the presence or absence of Nec-1s (10 μM) for indicated time. Cell death was measured as a function of time by SytoxGreen positivity assay (**b**, **d**). The levels of p-S166 RIPK1 and cleaved caspase-3 (CC3) were determined by immunoblotting (**c**, **e**). Liver sections from E14.5 embryos of indicated genotype were immunostained for p-S166 RIPK1 (**f**) and stained for HE (**g**). TUNEL assay (**h**) and CC3 IHC (**i**) were performed on E14.5 liver sections of indicated genotypes. **j** Quantitative RT-PCR analysis of the mRNA expression of cytokines and chemokines in E14.5 livers of indicated genotypes (*n* = 6). **k** Volcano plots showing fold-change and *P*-value for the comparison of E14.5 *Senp1⁻/⁻* fetal livers versus E14.5 *Senp1⁺/⁺* fetal livers (*n* = 3). Differentially expressed genes (fold change >1.5, *P* value <0.05) are depicted in red (upregulated) and blue (downregulated). **l** Quantitative RT-PCR analysis of the mRNA expression of *Ripk1* (*n* = 6). Heatmap (**m**) and gene ontology (GO) analysis (**n**) of 278 genes that are upregulated in E14.5 *Senp1⁻/⁻* fetal livers (vs *Senp1⁺/⁺* fetal livers) and downregulated in *Senp1⁻/⁻;Ripk1^D138N/D138N* fetal livers (vs *Senp1⁻/⁻* fetal livers) (*n* = 3). *n* = 8 embryos (**f–i**), *n* = 3 biologically independent experiments (**b–e**). Data are represented as mean ± s.d. (**b**, **d**), and mean ± s.e.m. (**h**, **i**, **j**, **l**). One-way ANOVA, post hoc Dunnett's test (**h**, **i**, **j**, **l**). Source data are provided in Source Data file.

receptor EPOR in *Senp1⁻/⁻;Ripk1^D138N/D138N* embryos were further confirmed by quantitative RT-PCR (Supplementary Fig. 5j). Thus, RIPK1 kinase-dead knockin mutation does not rescue the embryonic lethality of *Senp1⁻/⁻* mice probably due to the failure in preventing anemia caused by erythropoiesis defects and blood vessel development defects.

## SENP1 deficiency promotes RIPK1 activation and alters its ubiquitination in TNF-RSC

To characterize the mechanism by which SENP1 KD promotes RIPK1 activation and cell death in TNFα-stimulated cells, we next examined whether SENP1 regulates the activation of RIPK1 in TNF-RSC. We found that SENP1 KD dramatically enhanced the activation of RIPK1 in TNF-RSC (Fig. 4a). The activation of RIPK1 in TNF-RSC of SENP1-KD cells can be fully blocked by Nec-1s (Fig. 4b). Interestingly, SENP1 KD also increased RIPK1 ubiquitination (Fig. 4a), which cannot be blocked by the addition of Nec-1s (Fig. 4b). To characterize the mechanism by which SENP1 alters RIPK1 ubiquitination in TNF-RSC, we conducted a mass spectrometry analysis of TNF-RSC in SENP1-WT and KD MEFs stimulated by TNFα. We found that SENP1 KD increased the recruitment of LUBAC complex (including SHARPIN, HOIP, and HOIL), which performs M1-linked ubiquitination on RIPK1, while decreased that of A20, which removes K63-linked ubiquitination of RIPK1, in TNF-RSC after treating with TNFα for 5 min (Fig. 4c). Increased recruitment of SHARPIN, HOIP and decreased recruitment of A20 to TNF-RSC in TNFα-treated SENP1-KD MEFs were confirmed by immunoblotting (Fig. 4d). In contrast, SENP1 KD did not affect the recruitment of TRADD into TNF-RSC (Fig. 4c, d). Consistent with increased recruitment of LUBAC and decreased recruitment of A20 to TNF-RSC in SENP1-KD cells, both M1 and K63 ubiquitination of RIPK1 in TNF-RSC were increased (Fig. 4e). Alteration of RIPK1 ubiquitination patterns in TNF-RSC has been shown to promote RIPK1 activation[47,48]. For example, reduced A20 recruitment along with increased K63 ubiquitination of RIPK1 in TNF-RSC promotes RIPK1 activation in TNF-RSC[48,49]. Thus, the over-activation of RIPK1 in SENP1-KD cells likely attributed to the change in its ubiquitination patterns. We subsequently characterized the interaction of RIPK1 with FADD to form complex-II, a key downstream event. SENP1 KD led to an increased formation of complex II as compared to WT MEFs, and interaction of FADD and RIPK1 induced by TNFα in SENP1-KD MEFs was blocked by Nec-1s (Fig. 4f). Thus, SENP1 modulates the activation of RIPK1 by orchestrating its ubiquitination patterns in TNF-RSC, and SENP1 prevent TNFα-induced cell death by restricting RIPK1 activation and consequent complex-II formation.

## SUMOylation of RIPK1 in TNF-RSC is suppressed by SENP1

SENP1 deficiency alters RIPK1 ubiquitination and promotes its activation in TNF-RSC, which occurs within minutes after TNFα stimulation. This suggests that the substrate modulated by SENP1 may be a component of TNF-RSC, which plays an important role in controlling RIPK1 activation and cell death. To explore this possibility, we conducted a mass spectrometry analysis of SUMOylated proteins in WT and SENP1-KD MEFs stimulated by TNFα for 5 min (Supplementary Fig. 6a). We found that SENP1 KD increased the SUMOylation of proteins that are involved in TNFR1 signaling, including RIPK1, PPM1B, MAP2K2, NFKB2, and RelB, upon TNFα stimulation (Fig. 5a). Since RIPK1, but not the others, is a key component of TNF-RSC, we hypothesized that the loss of SENP1 leads to RIPK1 SUMOylation in TNF-RSC, hence altering RIPK1 ubiquitination and promoting its activation in TNF-RSC. To test our hypothesis, we first confirmed that RIPK1 was a substrate of SUMOylation by using in-cell SUMOylation assays in which exogenous RIPK1 was readily modified by SUMO3 in the presence of E2-conjugating enzyme Ubc9 (Fig. 5b).

Proteins modified by SUMOylation can be identified by in vitro reconstitution of SUMOylation assays using purified enzymes, substrate, and SUMO proteins in which E1 and E2 alone are sufficient for SUMO conjugation[21]. We then performed in vitro SUMOylation assays in which RIPK1 was purified from HEK293T cells and pretreated with pan-deubiquitylase USP2[50] to completely remove ubiquitination to avoid its interference with SUMOylation (Supplementary Fig. 6b). We found that RIPK1 can be directly SUMOylated in vitro in the presence of E1 (SAE1/UBA2) and E2 (Ubc9) (Fig. 5c). We next examined whether RIPK1 is SUMOylated in TNF-RSC in SENP1-KD cells after treating with TNFα. To this end, we performed a tandem immunoprecipitation (IP) of TNF-RSC to isolate SUMOylated RIPK1 in SENP1-WT and KD cells stably expressing a tagged SUMO3 which enables the enrichment of SUMOylated proteins. As shown in Fig. 5d, SENP1 KD significantly increased RIPK1 SUMOylation in TNF-RSC.

To examine whether RIPK1 is deSUMOylated by SENP1, we first characterized the interaction of SENP1 and RIPK1. We found that ectopically expressed RIPK1 was co-immunoprecipitated with SENP1 (Supplementary Fig. 6c). In contrast, A20 does not interact with SENP1 (Supplementary Fig. 6d). We also mapped the binding domain between SENP1 and RIPK1. We found that the intermediate domain (ID) of RIPK1 was mainly required for the interaction of RIPK1 and SENP1 (Supplementary Fig. 6e). Similarly, we found that the N-terminus, but not the C-terminus of SENP1 interacted with RIPK1 (Supplementary Fig. 6f). We next investigated whether SENP1 was recruited into TNF-RSC of TNFα-stimulated cells. We found that SENP1 was rapidly recruited into TNF-RSC within 5 min of TNFα stimulation (Fig. 5e). Consistent with the interaction of SENP1 with RIPK1, the recruitment of SENP1 into TNF-RSC was totally blocked in *Ripk1⁻/⁻* MEFs stimulated by TNFα (Fig. 5e).

Next, we examine whether SENP1 deSUMOylates RIPK1. In cell SUMOylation assays using nickel-nitrilotriacetic acid (Ni-NTA) purification showed that WT SENP1, but not its catalytically inactive mutant, significantly reduced the SUMOylation levels of exogenous RIPK1 (Supplementary Fig. 6g). To test whether SENP1 directly deSUMOylates RIPK1, we performed in vitro deSUMOylation assays in which SUMOylated RIPK1 was purified from HEK293T cells transfected with RIPK1 and SUMO3. We found that purified recombinant SENP1 efficiently removed SUMO from the SUMOylated RIPK1 (Fig. 5f). This result was also

confirmed by in vitro SUMOylation and deSUMOylation assays (Fig. 5g). Thus, RIPK1 is SUMOylated in TNF-RSC, which is suppressed by SENP1.

We next examined the SUMOylation levels of RIPK1 in human samples of NASH patients and that in the NAFLD mouse models that showed reduced SENP1 expression. We found that RIPK1 SUMOylation is substantially increased in the livers of individuals with NASH than that in the nonsteatotic controls (Supplementary Fig. 6h). Consistently, RIPK1 SUMOylation is also increased in the livers of mice with NAFLD than that in control mice (Supplementary Fig. 6i).

### RIPK1 is SUMOyalted at K550 and SUMOylation of RIPK1 promotes its activation

To determine the relevant lysine(s) that can be modified by SUMO in the RIPK1 protein, we conducted a mass spectrometry analysis of SUMOylated RIPK1 by using in cell SUMOylation assays, in which T90R-mutated SUMO3 was exogenously expressed to promote modified

substrates that when digested with trypsin produce remnants of two amino acids that can be detected by MS to identify SUMO conjugation sites[51] (Supplementary Fig. 7a, b). We found that overexpressing SUMO3-T90R and Ubc9 induced the SUMOylation of RIPK1 at multiple lysine residues with a conserved Lys550 (K550, mouse RIPK1) and K132 residues being the main SUMOylation sites of RIPK1 (Fig. 6a and Supplementary Fig. 7c). K550, but not K132, conforms to the SUMO modification consensus motif ($\psi$-K-×E/D)[51], which is also conserved among species (Fig. 6a). Mutation of K550 to Ala reduced SUMOylation of exogenously expressed RIPK1 (Supplementary Fig. 7d). In contrast, mutation of both K550 and K132 to Ala did not offer additional reduction of SUMOylation levels of RIPK1 (Supplementary Fig. 7d), thus RIPK1 is dominantly SUMOylated at K550.

To address a requirement for the SUMOylation of RIPK1 in controlling its activity and cell death, we reconstituted RIPK1 and SENP1 double-deficient MEFs with WT RIPK1 and K550A mutant RIPK1,

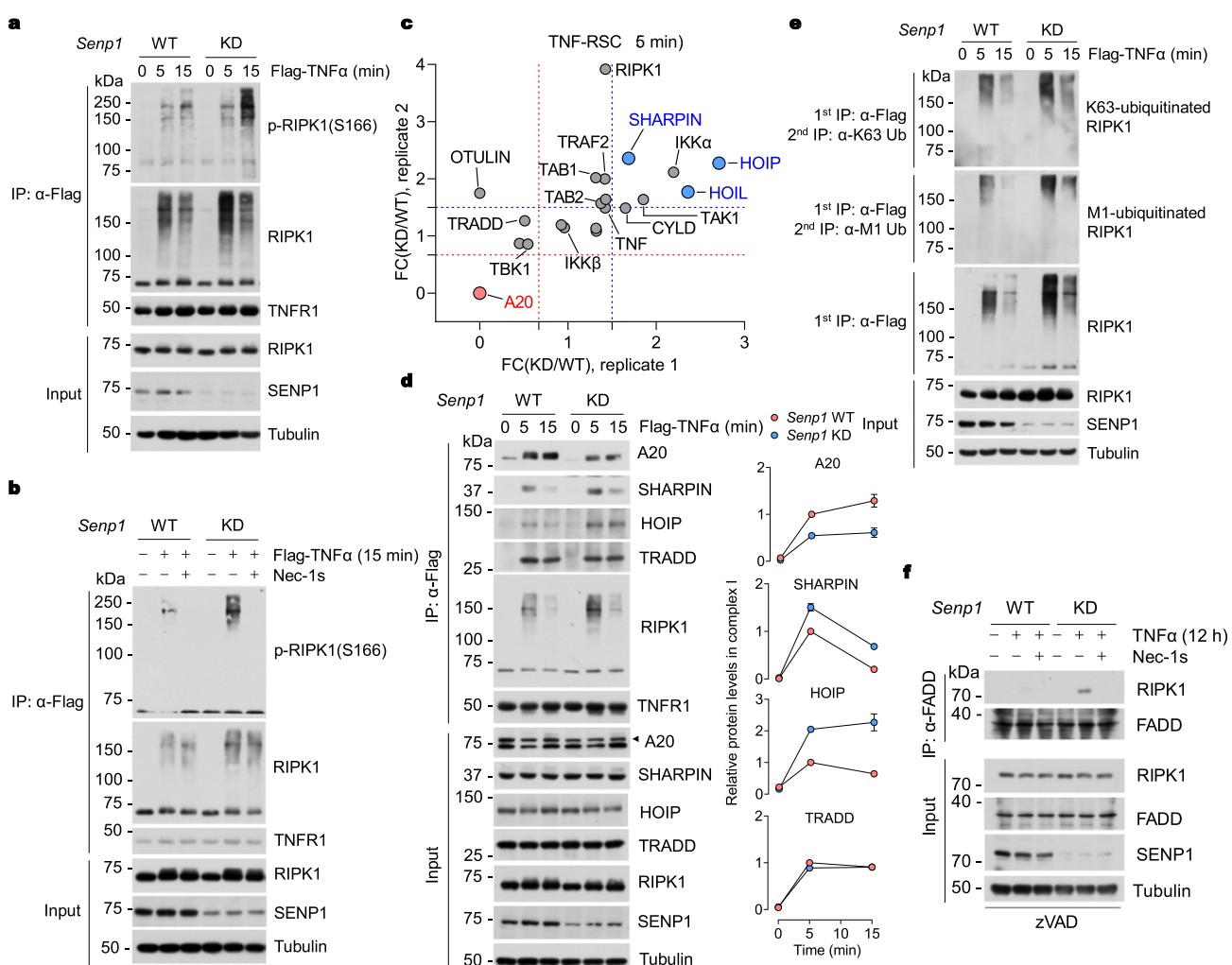

**Fig. 4 | SENP1 deficiency promotes RIPK1 activation and alters its ubiquitination in TNF-RSC.** MEFs were stimulated by Flag-TNFα (100 ng/ml) without (**a**) or with (**b**) Nec-1s (10 μM) for indicated time. TNF-RSC was immunoprecipitated using anti-Flag resin. The immune complexes were analyzed by immunoblotting using anti-p-S166 RIPK1 antibody and other antibodies as indicated. **c** Mass spectrometry analysis of TNF-RSC in *Senp1*-WT and *Senp1*-KD MEFs. TNF-RSC from Flag-TNFα (100 ng/ml, 5 min)-treated *Senp1*-WT and *Senp1*-KD MEFs was isolated and analyzed by mass spectrometry. The hits were selected based on the KD/WT fold change (FC) of the duplicate experiments with cut-offs set at FC(KD/WT) > 1.5 or <0.67. A20 (top hit) was labeled in red, LUBAC complex (top hits, HOIP, HOIL, and SHARPIN) were labeled in blue, and others were labeled in gray. **d** MEFs were stimulated by Flag-TNFα (100 ng/ml) for indicated time. TNF-RSC was immunoprecipitated using

anti-Flag resin. The immune complexes were analyzed by immunoblotting using anti-A20 antibody and other antibodies as indicated. Quantification of indicated protein in TNF-RSC was shown on the right. **e** MEFs were treated with Flag-TNFα (100 ng/ml) for indicated time. TNF-RSC was isolated by anti-Flag resin and denatured in 6 M urea. The TNF-RSC was further analyzed by immunoprecipitation using anti-M1 (6 M urea) or anti-K63 (3 M urea) ubiquitin antibody under denatured condition. The levels of RIPK1 ubiquitination were analyzed by immunoblotting. **f** MEFs were pre-incubated with zVAD.fmk (10 μM) in the presence or absence of Nec-1s (10 μM) for 0.5 h and then stimulated with 10 ng/ml TNFα for 12 h. The complex II was isolated by FADD immunoprecipitation and RIPK1 binding was revealed by immunoblotting. $n = 3$ biologically independent experiments (**a, b, d–f**). Data are represented as mean ± s.e.m. (**d**). Source data are provided in Source Data file.

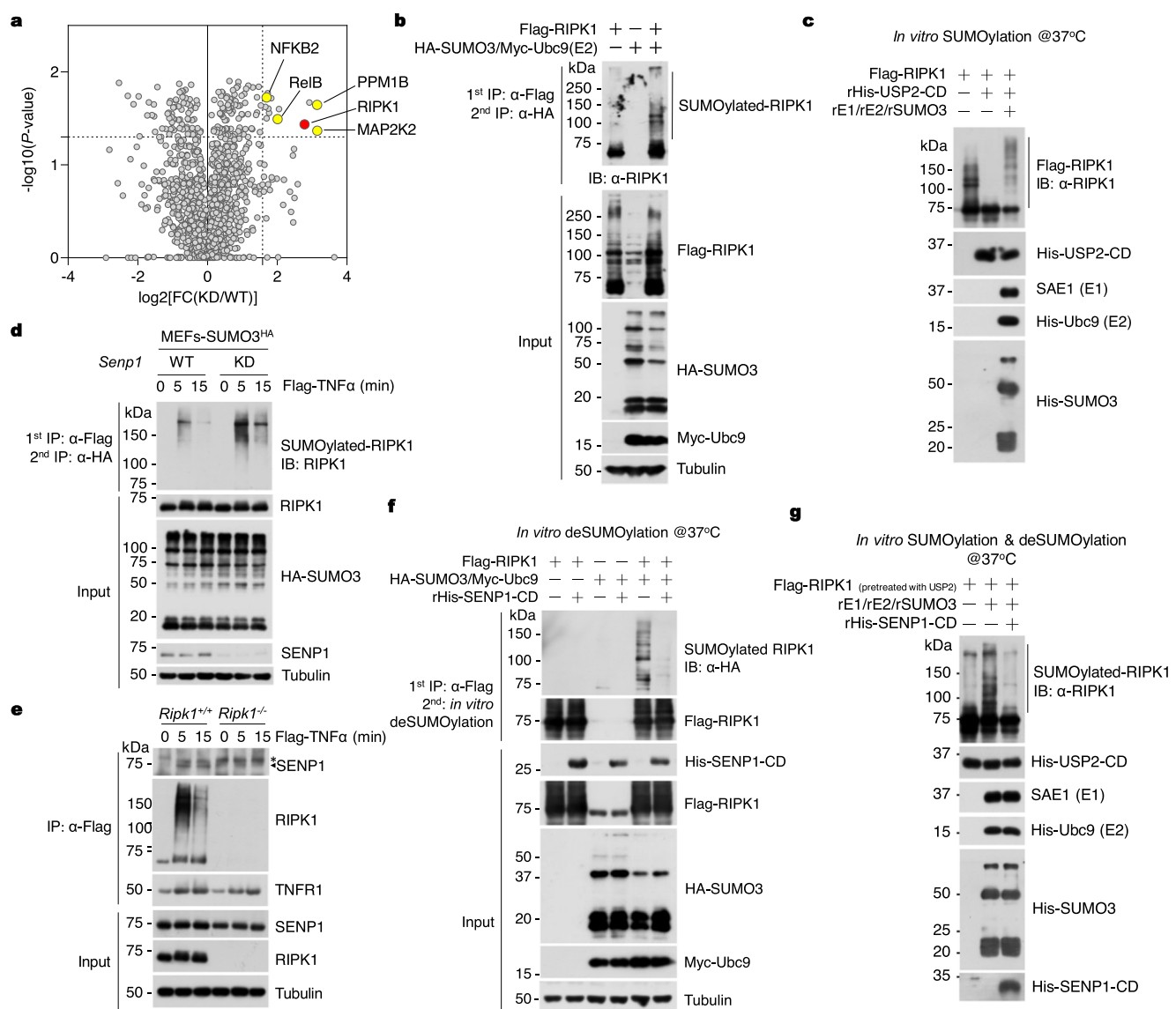

**Fig. 5 | SENP1 is recruited into TNF-RSC to suppress RIPK1 SUMOylation. a** Mass spectrometry analysis of SUMOylated proteins in SENP1-KD cells after TNFα stimulation. MEFs were stimulated by 10 ng/ml TNFα for 5 min. SUMOylated proteins were isolated using anti-HA resin followed by mass spectrometry analysis. The hits were selected based on the KD/WT fold change (FC) of the triplicate experiments with cut-offs set at log2[FC(KD/WT)] > 1.58 and *P* value <0.05. Yellow, SUMOylation substrates that are involved in TNFα signaling. Red, RIPK1. Unpaired two-tailed t-test. **b** HEK293T cells were transfected with Flag-RIPK1, HA-SUMO3 and Myc-Ubc9 for 24 h. SUMO3-modified RIPK1 was enriched by tandem immunoprecipitation as indicated, and then analyzed by immunoblotting using anti-RIPK1 antibody. **c** Flag-RIPK1 was isolated from HEK293T cells expressing this construct using anti-Flag resin and pre-treated with the catalytic domain of USP2 (USP2-CD) for 0.5 h followed by in vitro SUMOylation assay in the presence of E1 (SAE1/UBA2), E2 (Ubc9) and SUMO3 as indicated for 1 h. The samples were then analyzed by immunoblotting with anti-RIPK1 antibody. **d** MEFs were stimulated by Flag-TNFα

(100 ng/ml) for indicated time. SUMOylated RIPK1 was isolated by tandem immunoprecipitation of TNF-RSC, and then analyzed by immunoblotting with anti-RIPK1 antibody. **e** MEFs were stimulated by 100 ng/ml Flag-TNFα for the indicated time. TNF-RSC was immunoprecipitated using anti-Flag resin. The recruitment of SENP1 into TNF-RSC was analyzed by immunoblotting with anti-SENP1 antibody. *: non-specific band. **f** SUMOylated RIPK1 was isolated from HEK293T cells expressing constructs encoding Flag-RIPK1, HA-SUMO3, and Myc-Ubc9 followed by in vitro deSUMOylation assay in the presence of the catalytic domain of SENP1 (SENP1-CD) as indicated for 4 h. The samples were then analyzed by immunoblotting with anti-HA antibody. **g** Flag-RIPK1 was isolated from HEK293T cells expressing this construct and pre-treated with USP2-CD followed by in vitro SUMOylation assay in the presence of E1, E2, and SUMO3. The samples were then incubated with SENP1-CD for 4 h, and analyzed by immunoblotting with anti-RIPK1 antibody. *n* = 3 biologically independent experiments (**b**–**g**). Source data are provided in Source Data file.

respectively (Supplementary Fig. 7e). We found that the SUMOylation levels of K550A RIPK1 in TNF-RSC were largely reduced compared with that of WT RIPK1 in TNFα-stimulated SENP1-KD cells (Fig. 6b). K550A mutation of RIPK1 also reduced the ubiquitination levels of RIPK1 and the levels of p-S166 RIPK1 in TNF-RSC compared with that of WT RIPK1 in TNFα-stimulated SENP1-KD cells (Fig. 6c). Moreover, disruption of RIPK1 SUMOylation by K550A mutation increased the recruitment of A20, while decreased the recruitment of LUBAC into TNF-RSC in TNFα-

stimulated SENP1-KD cells (Fig. 6d and Supplementary Fig. 7f). Reconstitution of WT RIPK1 in RIPK1-KO;SENP1-KD cells restored their sensitivity to TNFα-induced cell death, while reconstitution of K550A RIPK1 in RIPK1-KO;SENP1-KD cells abolished TNFα-induced cell death (Fig. 6e). Consistently, disruption of RIPK1 SUMOylation by K550A mutation in SENP1-KD cells inhibited RIPK1 activation and the cleavage of caspase-3 (Fig. 6f). Thus, SUMOylation of RIPK1 is essential for orchestrating its activation in TNF-RSC in SENP1-KD cells.

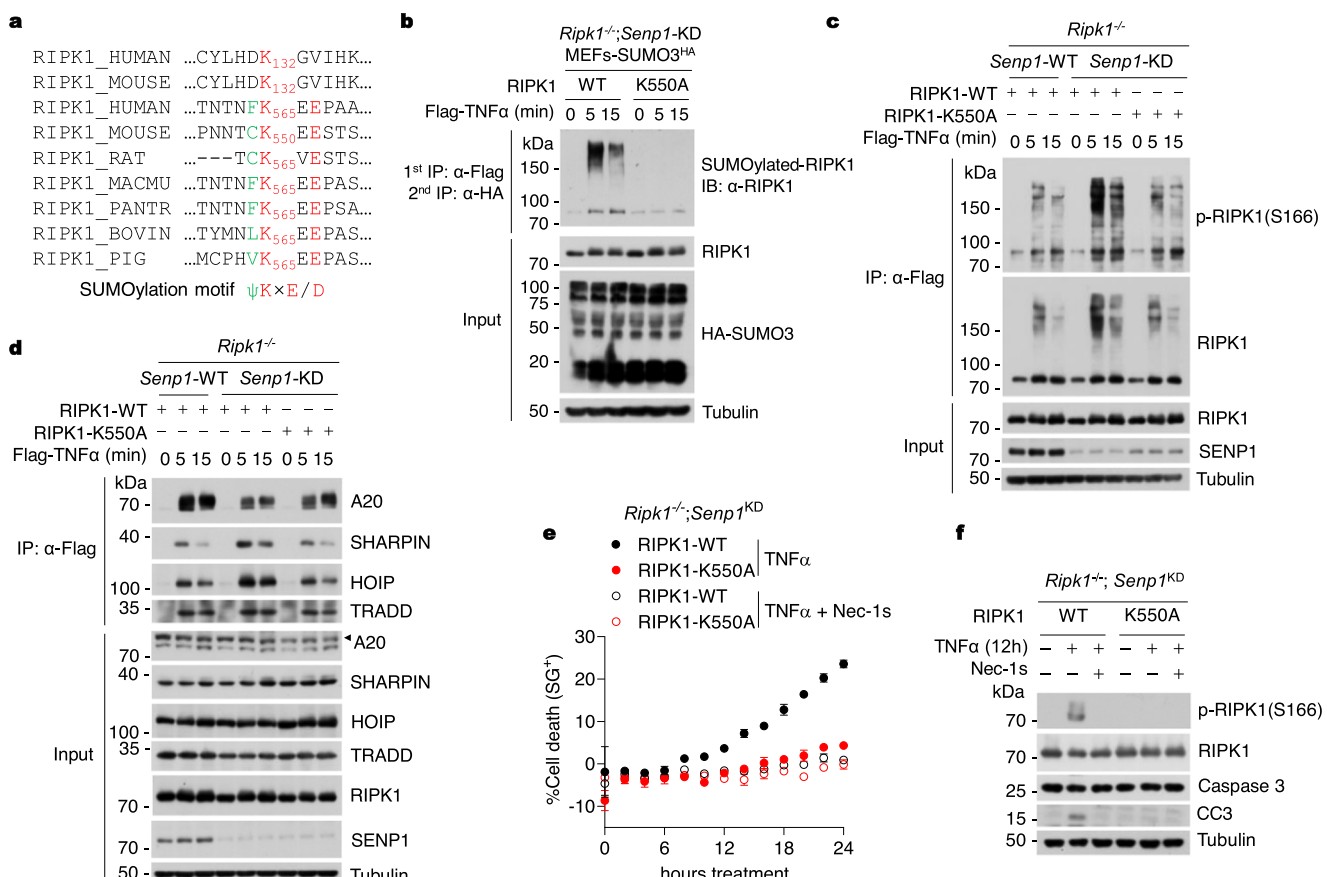

**Fig. 6 | RIPK1 is SUMOyalted at K550 and SUMOylation of RIPK1 promotes its activation. a** Multiple sequence alignment of RIPK1 proteins from different species. K550 of murine RIPK1 (K565 in human RIPK1) locates within SUMO modification consensus motif (ψ-K-×E/D), which is highly conserved among different species, but not that of K132 of RIPK1. ψ: hydrophobic amino acid; K: the target lysine; ×: any amino acid; E or D is an acidic residue. **b** MEFs were stimulated by Flag-TNFα (100 ng/ml) for indicated time. SUMOylated RIPK1 was isolated by tandem immunoprecipitation of TNF-RSC, and then analyzed by immunoblotting with anti-RIPK1 antibody. MEFs were stimulated by Flag-TNFα (100 ng/ml) for indicated time. TNF-RSC was immunoprecipitated using anti-Flag resin. The immune complexes were analyzed by immunoblotting using anti-RIPK1 antibody and anti-p-S166 RIPK1 antibody (**c**), and anti-A20 antibody and other antibodies as indicated (**d**). MEFs were treated with TNFα (10 ng/ml) in the presence or absence of Nec-1s (10 μM). Cell death was measured as a function of time by SytoxGreen positivity assay (**e**). The level of p-S166 RIPK1 and CC3 were determined by immunoblotting (**f**). *n* = 3 biologically independent experiments (**b–f**). Data are represented as mean ± s.d. (**e**). Source data are provided in Source Data file.

## RIPK1 is SUMOylated by PIAS1 in SENP1-deficient cells

We next characterized the SUMO E3 ligase that mediates the SUMOylation of RIPK1 in SENP1-deficient conditions. Since knockdown of PIAS1 prevented TNFα/5z7-induced RDA (Fig. 3a), we investigated the role of PIAS1 as a positive regulator of RIPK1-dependent cell death. PIAS1 has been implicated in the regulation of NF-κB pathway and restricting inflammation[52]. *Pias1[−/−]* mice appear normal and show increased protection against pathogenic infections[53]. The inflammation-repressing activity of PIAS1 is mediated by its association with IKKα, a component of TNF-RSC, and the phosphorylation on Ser90 residue by IKKα within minutes after TNFα stimulation[54]. We reasoned that PIAS1 might be the SUMO E3 ligase mediating the SUMO modification of RIPK1 in SENP1-deficient cells. To this end, we first tested whether PIAS1 was associated with RIPK1. We found that PIAS1 was present in RIPK1 immunoprecipitates when co-expressing PIAS1 and RIPK1 in HEK293T cells (Supplementary Fig. 8a). We also mapped the binding domain between PIAS1 and RIPK1. Similar to SENP1, we found that PIAS1 also interacted with the intermediate domain (ID) of RIPK1 (Supplementary Fig. 8b).

To examine whether PIAS1 SUMOylates RIPK1, we performed in cell SUMOylation assays using Ni-NTA purification, which showed that PIAS1 significantly increased the SUMOylation of exogenous RIPK1 (Supplementary Fig. 8c). We next tested whether the SUMOylation of RIPK1 in TNF-RSC of TNFα-stimulated SENP1-KD cells was mediated by PIAS1, we performed a tandem IP of TNF-RSC to isolate SUMOylated RIPK1 in SENP1 and PIAS1 double knockdown cells (Supplementary Fig. 8d). We found that PIAS1 knockdown reduced RIPK1 SUMOylation in TNF-RSC of SENP1-KD cells that were stimulated with TNFα (Supplementary Fig. 8e).

We next investigated whether PIAS1 was recruited into TNF-RSC of TNFα-stimulated cells. We found that PIAS1 was also recruited to the TNF-RSC within minutes of TNFα stimulation (Supplementary Fig. 8f). Consistent with the interaction of PIAS1 and RIPK1, the recruitment of PIAS1 into TNF-RSC was totally blocked in *Ripk1[−/−]* MEFs stimulated by TNFα (Supplementary Fig. 8f). We further explored the recruitment of SENP1 in PIAS1-KO cells, and the recruitment of PIAS1 in SENP1-KO cells to determine the sequence of PIAS1 and SENP1 recruitment in TNF-RSC. We found that the recruitment of SENP1 in TNF-RSC is reduced in PIAS1-KO cells when compared to that in PIAS1-WT cells (Supplementary Fig. 8g), while the recruitment of PIAS1 is not altered in SENP1-KO cells (Supplementary Fig. 8h), suggesting that PIAS1 is at least partially required for the recruitment of SENP1. Because SENP1 is known to interact with SUMO2 noncovalently, which is necessary for its deSUMOylation activity[55], PIAS1 might be recruited first to perform SUMOylation on RIPK1, then the SUMO chains on RIPK1 increased SENP1 recruitment.

To demonstrate the role of PIAS1 in RIPK1 SUMOylation in SENP1-KD cells, we assessed the influence of PIAS1 on TNFα-induced RDA in SENP1-KD cells. We found that PIAS1 knockdown dramatically reduced cell death in SENP1-KD cells that were treated with TNFα (Supplementary Fig. 8i). PIAS1 knockdown also inhibited RIPK1 activation and the levels of CC3 (Supplementary Fig. 8j). Thus, PIAS1 increases RIPK1 SUMOylation in TNF-RSC of SENP1-deficient cells, which promotes RIPK1 activation and cell death. Taken these results together, we conclude that modulation of RIPK1 SUMOylation by PIAS1 and SENP1 in TNF-RSC may play a critical role in regulating ubiquitination and activation of RIPK1.

## SENP1 deficiency exaggerates LPS-induced acute liver injury in a RIPK1-dependent manner

Our results suggested that SENP1 inhibits RIPK1 activation and cell death induced by TNFα stimulation. Because TNFα is highly upregulated in lipopolysaccharides (LPS)-induced acute liver injury (ALI), we then tested if SENP1 plays a role in LPS-induced ALI. We used 2 months old mice, when spontaneous liver damage are not observed in *Senp1*^*f/f*;*Alb-Cre* mice (Supplementary Fig. 9a). After LPS injection for 4 h, we observed substantial liver damage in *Senp1*^*f/f*;*Alb-Cre* mice compared to *Senp1*^*f/f* littermate controls as determined by ALT, AST levels and histology stains (Supplementary Fig. 9b, c). Consistently, *Senp1*^*f/f*;*Alb-Cre* livers displayed increased number of TUNEL+ and p-S166 RIPK1+ cells compared to *Senp1*^*f/f* livers after LPS stimulation (Supplementary Fig. 9d). In addition, the mRNA levels of inflammatory cytokines, including *Tnf*, *Il1a*, *Il1b*, *Il6*, *Ccl2*, and *Ccl5* were substantially higher in *Senp1*^*f/f*;*Alb-Cre* livers than that in *Senp1*^*f/* livers (Supplementary Fig. 9e). Strikingly, RIPK1-D138N mutation largely reduced the liver damage and inflammation in *Senp1*^*f/f*;*Alb-Cre*;*Ripk1*^*D138N/D138N* mice (Supplementary Fig. 9b–e). Taken together, these results demonstrated the essential role of RIPK1 kinase activity in mediating liver inflammation and damage in LPS-induced ALI, under a hepatocyte-specific SENP1-deficiency condition.

## RIPK1 inactivation inhibits NAFLD induced by a high-fat diet in SENP1-deficient mice

We next investigated the function of SENP1-mediated RIPK1 deSU-MOylation in the pathogenesis of NAFLD. In primary mouse hepatocytes, SENP1 deficiency substantially increased RIPK1 SUMOylation levels after exposure to palmitic acid (PA), which \was performed to mimic in vivo NAFLD pathologies (Supplementary Fig. 9f). We also observed substantial cell death in SENP1-deficient hepatocytes treated with PA (Supplementary Fig. 9g). Consistently, RIPK1 activation as determined by its S166 phosphorylation, and apoptosis as shown by increased levels of CC3 are increased in *Senp1*^*f/f*;*Alb-Cre* hepatocytes treated with PA (Supplementary Fig. 9h). In addition, the apoptosis of *Senp1*^*f/f*;*Alb-Cre* hepatocytes induced by PA was fully protected by RIPK1-D138N mutation (Supplementary Fig. 9g, h). These results suggest that SENP1 deficiency in hepatocytes promotes RIPK1 SUMOylation, activation, and apoptosis in response to PA in vitro.

We further investigated the role of SENP1-mediated RIPK1 suppression in the pathogenesis of NAFLD by feeding mice with HFD for 16 weeks to induce moderate hepatic steatosis, hepatic inflammation, liver damage, and fibrosis[18]. *Senp1*^*f/f*;*Alb-Cre* mice were identical to *Senp1*^*f/f* mice with respect to body weight and liver weight after feeding with HFD (Fig. 7a). We found a significant increase of serum levels of ALT and AST, suggesting exaggerated liver damage, in HFD-fed *Senp1*^*f/f*;*Alb-Cre* mice compared with that of *Senp1*^*f/f* control mice (Fig. 7b). Strikingly, RIPK1-D138N mutation markedly reduced the serum levels of ALT and AST in HFD-fed *Senp1*^*f/f*;*Alb-Cre*;*Ripk1*^*D138N/D138N* mice compared with that of *Senp1*^*f/f*;*Alb-Cre* mice (Fig. 7b). TUNEL staining demonstrated that knockout of SENP1 substantially increased the number of apoptotic cells in the liver of HFD-fed *Senp1*^*f/f*;*Alb-Cre* mice, which can be prevented by inhibition of RIPK1 kinase (Fig. 7c).

H&E staining revealed that the hepatic steatosis and lipid accumulation were markedly aggravated in HFD-fed *Senp1*^*f/f*;*Alb-Cre* mice compared with that of HFD-fed *Senp1*^*f/f* mice (Fig. 7d), which was further confirmed by ORO staining and measuring hepatic concentrations of triglyceride (TG) and total cholesterol (TC) (Fig. 7e and Supplementary Fig. 9i). In comparison, the hepatic steatosis and lipid accumulation in the liver of HFD-fed *Senp1*^*f/f*;*Alb-Cre*;*Ripk1*^*D138N/D138N* mice were considerably lower (Fig. 7d, e and Supplementary Fig. 9i). HFD-fed *Senp1*^*f/f*;*Alb-Cre* mice also showed increased liver macrophage infiltration compared with HFD-fed *Senp1*^*f/f* mice, which was suppressed by RIPK1 D138N mutation (Fig. 7f). In keeping with this observation, the mRNA levels of multiple pro-inflammatory cytokines and chemokines were significantly higher in the livers of HFD-fed *Senp1*^*f/f*;*Alb-Cre* mice than in HFD-fed *Senp1*^*f/f* mice, and this difference disappeared in the livers of HFD-fed *Senp1*^*f/f*;*Alb-Cre*;*Ripk1*^*D138N/D138N* mice (Fig. 7g).

Masson's trichrome staining of liver tissue sections showed that HFD-fed *Senp1*^*f/f*;*Alb-Cre* mice developed higher levels of pericellular fibrosis compared with HFD-fed *Senp1*^*f/f* mice, which virtually disappeared in the livers of HFD-fed *Senp1*^*f/f*;*Alb-Cre*;*Ripk1*^*D138N/D138N* mice (Fig. 7h). We further verified this result by examining the expression of fibrogenic genes using quantitative RT-PCR. This analysis confirmed that the mRNA levels of fibrogenic parameters were increased in HFD-fed *Senp1*^*f/f*;*Alb-Cre* mice but was restored to the WT levels in HFD-fed *Senp1*^*f/f*;*Alb-Cre*;*Ripk1*^*D138N/D138N* mice (Fig. 7i). Thus, liver-specific SENP1 knockout exaggerates hepatic steatosis, inflammation, fibrosis, and liver damage in an HFD-induced NAFLD mouse model in a RIPK1 kinase-dependent manner.

## Discussion

TNF-RSC provides a critical checkpoint in TNFα signaling pathway that decides if RIPK1 kinase is to be activated. Various post-translational modifications of RIPK1 are at the core of this regulation[6,15]. Many key regulators of RIPK1, such as A20, LUBAC, NEMO, TAK1, and IKKα/β, are also recruited into TNF-RSC rapidly to directly regulate the activation of RIPK1 by modulating its ubiquitination and phosphorylation patterns[6]. RIPK1 can also undergo SUMOylation upon etoposide treatment and this modification seems to be required for optimal NF-κB activation after DNA damage[56]. However, RIPK1 SUMOylation and its effects in response to TNFα stimulation are unknown. In this study, we demonstrated that SENP1 is recruited into TNF-RSC by RIPK1 to limit SUMO modification on RIPK1 in TNFα-stimulated cells. RIPK1 is SUMOylated at K550 in TNF-RSC in SENP1-deficient cells in response to TNFR1 ligation. We showed that K550 SUMOylation of RIPK1 critically regulates its activation by orchestrating TNF-RSC and modulating its ubiquitination patterns in SENP1-deficient cells (Fig. 8). Our study demonstrates that SENP1-mediated deSUMOylation of RIPK1 is an essential checkpoint in TNF-RSC that controls RIPK1 kinase and RIPK1-driven cell death and inflammation. Accordingly, genetic ablation of SENP1 resulted in TNFα-induced cell death, both in cell lines and in vivo, as a consequence of unleashed RIPK1 activity.

RIPK1 kinase plays an important role in driving liver inflammation, and hepatocellular death. Hepatocyte-specific deletion of many known regulators of RIPK1 kinase, such as NEMO, and TAK1, leads to hepatocellular death, steatohepatitis, fibrosis, and hepatocellular carcinoma (HCC)[57,58]. Expression of a kinase-inactive RIPK1-D138N mutant in mice lacking NEMO or TAK1 in liver significantly ameliorated the observed liver pathology by reducing hepatocyte apoptosis and liver inflammation. As a result, steatohepatitis and fibrosis were decreased and HCC development was prevented[57,58]. In this study, we discovered that SENP1 is progressively decreased in proportion to NASH severity. We showed that hepatocyte-specific loss of SENP1 drives spontaneous NASH-related phenotypes, including liver inflammation, hepatocellular death, fibrosis, and steatosis. We further showed that RIPK1-D138N mutation in liver-specific *Senp1* knockout mice prevented all

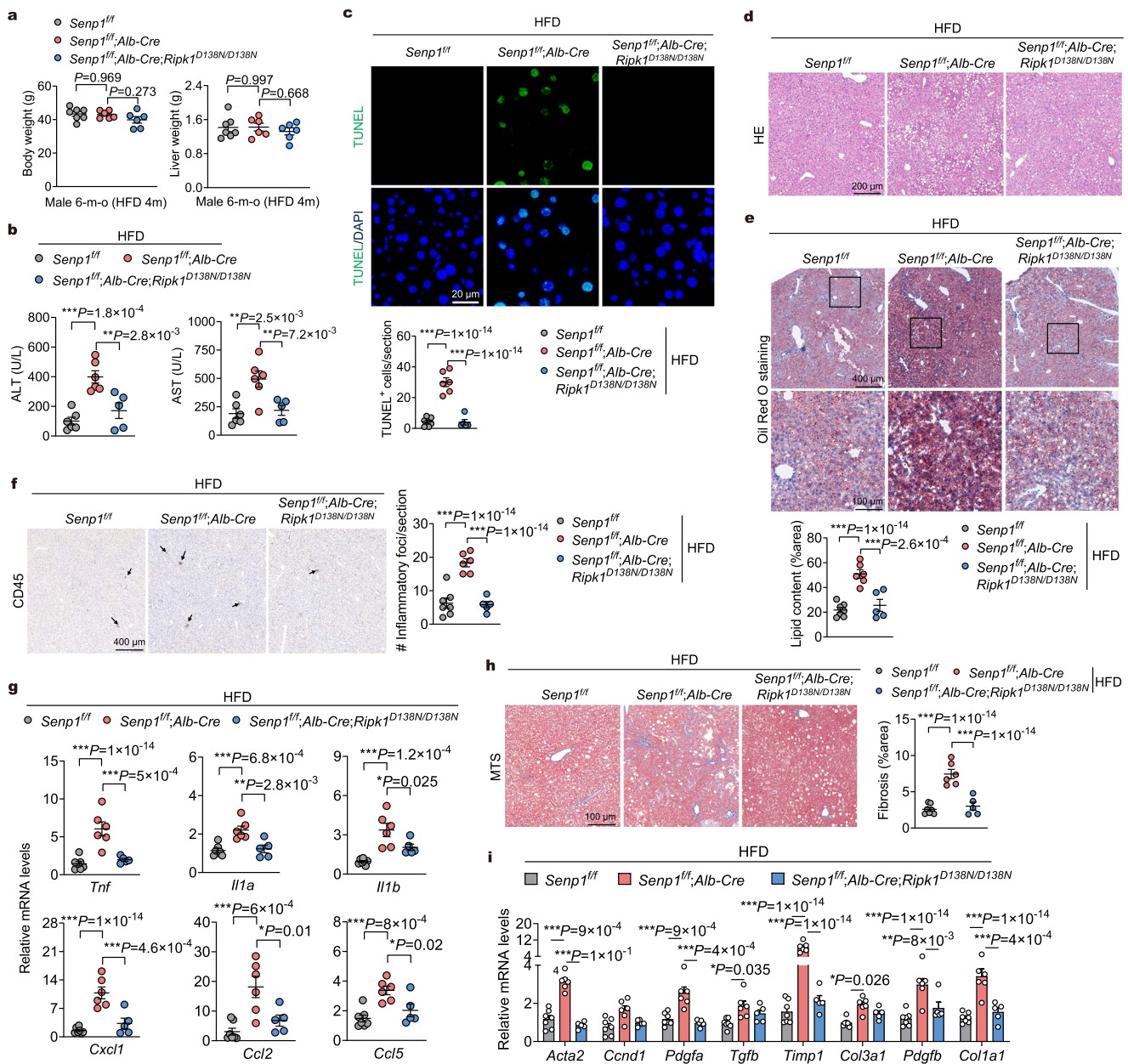

**Fig. 7 | Hepatic SENP1 deficiency aggravates RIPK1-driven liver damage in HFD-induced NAFLD. a** Body and liver weight of 6 months old male mice with indicated genotypes after feeding with high-fat-diet (HFD) for 4 months (*n* = 7 for *Senp1*^f/f^, *n* = 6 for *Senp1*^f/f^;*Alb-Cre* and *Senp1*^f/f^;*Alb-Cre*;*Ripk1*^D138N/D138N^). **b** Serum levels of ALT and AST of mice from **a** (*n* = 6 for *Senp1*^f/f^ and *Senp1*^f/f^;*Alb-Cre*, *n* = 5 for *Senp1*^f/f^;*Alb-Cre*;*Ripk1*^D138N/D138N^). **c** TUNEL assay was performed on liver sections of mice from **a**. Graph depicting numbers of TUNEL^+^ cells on sections of indicated genotypes. HE staining (**d**) and Oil Red O staining (**e**) of liver sections of mice from **a**. Graph depicting percentage of lipid content on liver sections of indicated genotypes (**e**). **f** Immunohistochemistry of CD45 of liver sections of mice from **a**. Arrows indicates

infiltration of macrophages. Graph depicting numbers of CD45^+^ foci on liver sections of indicated genotypes. **g** Quantitative RT-PCR analysis of the mRNA expression of cytokines and chemokines in livers of mice from **a**. **h** Representative images of Masson's trichrome stained (MTS) liver sections of mice from **a**. Graph depicting percentage of fibrosis area on liver sections of indicated genotypes. **i** Quantitative RT-PCR analysis of the mRNA expression of fibrogenic parameters in livers of mice from **a**. *n* = 7 mice for *Senp1*^f/f^, *n* = 6 mice for *Senp1*^f/f^;*Alb-Cre*, and *n* = 5 mice for *Senp1*^f/f^;*Alb-Cre*;*Ripk1*^D138N/D138N^ (**c–i**). Data are represented as mean ± s.e.m. (**a–c**, **e–i**). One-way ANOVA, post hoc Dunnett's test (**a–c**, **e–i**). Source data are provided as a Source Data file.

these phenotypes observed, suggesting an important physiological role of SENP1-mediated RIPK1 suppression in maintaining liver homeostasis. Moreover, hepatocyte-specific loss of SENP1 exaggerates liver damage in HFD-induced NAFLD, which is suppressed by RIPK1-D138N mutation. Since liver inflammation and hepatocellular death are underlying mechanisms in promoting NASH, we suggest the possibility of inhibiting RIPK1 kinase as a therapeutic strategy for reducing NASH pathogenesis by inhibiting liver inflammation and hepatocellular death. In addition, due to sex differences in susceptibility to NAFLD[59],

only male mice were used in HFD-induced NAFLD models. Future studies will incorporate females to determine potential sex differences in these effects.

Dysregulation of RIPK1 signaling is also involved in a heterogeneous group of monogenic immune and autoinflammatory diseases. Many of these disease-associated genes are involved in regulating both NF-κB signaling and RIPK1 activation, including A20, ABIN1, NEMO, OTULIN and LUBAC complex[5,6,60]. Genetic studies suggested some amount of disease pathology in humans can be attributed to aberrant

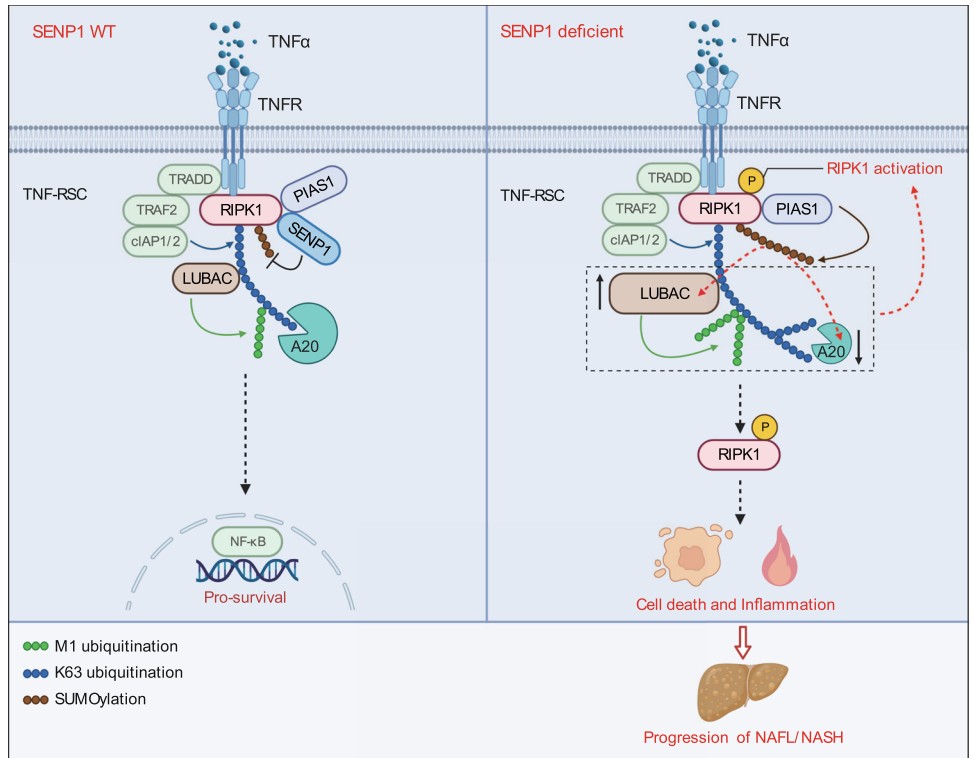

**Fig. 8 | A model: SENP1 limits RIPK1 activation by deSUMOylation in NASH.** In control condition (left), SENP1 is recruited into TNF-RSC in response to TNFα in a manner dependent on RIPK1 to deSUMOylate RIPK1 in TNF-RSC and put RIPK1 in check. In SENP1 deficient condition (right), RIPK1 is hyper-SUMOylated in TNF-RSC. The SUMOylation of RIPK1 can be enhanced by PIAS1, which was also recruited by RIPK1 in TNF-RSC. Increased RIPK1 SUMOylation re-orchestrates TNF-RSC by reducing the recruitment of A20 and increasing that of LUBAC into TNF-RSC, which results in both increased K63 and M1 ubiquitination of RIPK1. Altered RIPK1 ubiquitination patterns in TNF-RSC thus promote RIPK1 activation, and license cells to die through RIPK1 kinase-dependent cell death in response to TNFα, leading to progression of NASH. Created with BioRender.com.

RIPK1 regulation. Similar to these genes, adipocyte-specific deletion of SENP1 also leads to autoimmune-mediated damages, which drives type 1 diabetic phenotype in mice[30]. Interestingly, low-penetrance coding and non-coding variants in A20 have been also suggested to underlie type 1 diabetes mellitus (T1DM)[61]. We propose that SENP1 deficiency-induced RIPK1 desuppression might be important as a mechanism to drive T1DM. Thus, exogenous triggers that lead to transient inflammation in healthy subjects may promote sustained inflammation and cell death involving different tissues and organs in individuals with aberrant RIPK1 regulation.

SUMOylation is different from ubiquitination in terms of E3 ligases involved. An E3 ligase is non-essential for SUMOylation but does provide precision and efficacy for SUMO conjugation[62]. In this study, we showed that the SUMO E3 ligase PIAS1 is important for RIPK1 SUMOylation in SENP1-deficient condition. PIAS1 enhanced RIPK1 SUMOylation and its activation in TNFα-stimulated SENP1-KD cells. Similar to SENP1, PIAS1 is also recruited into TNF-RSC by RIPK1 upon TNFα stimulation, which enables its performing SUMOylation on RIPK1. PIAS1 has been shown to play an important role in NF-κB signaling. PIAS1 negatively regulates NF-κB signaling by blocking the binding of NF-κB to gene promoters[52]. Intriguingly, PIAS1 was previously shown to be rapidly phosphorylated by IKKα in response to TNFα stimulation, which is required for the ability of PIAS1 to block the promoter binding of NF-κB p65 subunit to restrict inflammation and immunity[54]. Our results provide an explanation for this observation, as the recruitment of PIAS1 into TNF-RSC might enable its phosphorylation by IKKα. Our study defines a molecular contribution of PIAS1 that promotes the activity of RIPK1 kinase in mediating apoptosis in SENP1-deficient cells, which is distinct from

the role of PIAS1 in limiting inflammation in canonical NF-κB pathway. Such difference can be observed in other regulators of both NF-κB pathway and the kinase activity of RIPK1. For example, TAK1 is a positive regulator of NF-κB-dependent inflammation[63], and also an endogenous inhibitor of RIPK1 kinase activity[64]. Ablation of TAK1 leads to suppress of canonical NF-κB-mediated inflammatory response, but promotes RIPK1 kinase-dependent apoptosis due to unleashed RIPK1 activation[64].

We herein identified a crucial role for SENP1 in preventing TNFα-induced cell death. So far, SENP1 has mainly been considered as a modulator of gene expression, mostly in GATA1/2-dependent gene expression, which regulate erythropoiesis and vascular development, as well as in NEMO-dependent NF-κB pathway[28–30]. However, SENP1 deficiency in mice leads to not only erythropoiesis defects and vascular development defects but also massive fetal liver apoptosis, resulting in embryonic lethality. Our results suggest that the liver cell death phenotype, but not erythropoiesis defects and vascular development defects, is mainly attributed to aberrant RIPK1 activation. Genetic inactivation of RIPK1 ameliorated the apoptotic phenotype of fetal liver of *Senp1*[−/−] mice, despite the failure in rescuing the embryonic lethality of *Senp1*[−/−] mice.

Taken together, we uncovered a SUMOylation-dependent mechanism underlying RIPK1 activation during the development of NASH. We discovered that by deSUMOylating RIPK1 within TNF-RSC, SENP1 controls a pathophysiologically essential cell death checkpoint in TNFα signaling. The findings provide evidence supporting a role for SENP1 in hepatocytes as a robust suppressor of NASH and reveal promising avenue for the development of a new treatment for NASH in the clinic.

## Methods

### Ethics statement

Animal experiments were conducted according to the protocols approved by the Standing Animal Care Committee at the Interdisciplinary Research Center on Biology and Chemistry (Approval no. ECSIOC2021-08). All procedures involving human samples were approved by the Ethics Committee of Xinhua Hospital affiliated to Shanghai Jiao Tong University School of Medicine (Approval no. XHEC-D-2022-040), and were consistent with the principles outlined in the Declaration of Helsinki. Written informed consent was obtained from subjects or families of all participants.

### Animals

*Senp1*$^{+/-}$ mice, *Senp1*$^{flox/flox}$ mice were kindly provided by Jinke Cheng of Shanghai Jiao Tong University College of Basic Medical Sciences, China. *Alb-Cre* mice were from The Jackson Laboratory (Catalog No. 003574). *Ripk1*$^{D138N/D138N}$ mice were generated as previously reported[58]. All mice were in the C57BL/6 J background. *Senp1*$^{+/-}$ mice were crossed with *Ripk1*$^{D138N/D138N}$ mice to generate *Senp1*$^{+/-}$;*Ripk1*$^{D138N/D138N}$ mice. *Senp1*$^{flox/flox}$;*Alb-Cre* mice were crossed with *Ripk1*$^{D138N/D138N}$ mice to generate *Senp1*$^{flox/flox}$;*Alb*-Cre;*Ripk1*$^{D138N/D138N}$ mice. For high-fat-diet (HFD)-induced NAFLD mouse model, 8-week-old male mice of indicated genotypes were fed a normal diet (ND: 10% kcal fat, D12450J, Research Diets Inc.) or high-fat diet (HFD: 60 kcal% fat, D12492, Research Diets Inc.) for 16 weeks (the time necessary for an inflammatory and fibrotic response to develop). All animals were maintained in a specific pathogen-free environment and housed with no more than five animals per cage under controlled light (12 h light and 12 h dark cycle), temperature ($24 \pm 2$ °C), and humidity ($50 \pm 10$%) conditions, and provided with *ad libitum* access to food and water throughout all experiments. Local and federal regulations regarding animal welfare were followed. General welfare monitoring was performed on a daily basis. Prior to individual experiments, each animal was diligently checked for its suitability according to preset criteria approved by the local animal welfare authorities. Prior to liver harvest, mice were euthanized by isoflurane overdose followed by cervical dislocation. This is an approved method according to the recommendations of the panel on Euthanasia of the American Veterinary Medical Association.

### Human liver samples

Steatotic livers were obtained from individuals with NAFLD or NASH who underwent liver biopsy or steatotic liver transplantation. Liver steatosis due to excessive alcohol consumption (>140 g for men or >70 g for women, per week), use of toxins or drugs, and viral infection (for example, hepatitis B virus and hepatitis C virus) were excluded from the study. Samples from nonsteatotic liver were collected from the normal donor livers. All donor's livers were allocated via China Organ Transplant Response System from 2017 to 2021. The donors were enrolled in the study on a volunteer basis, and the families of organ donors were approached for consent. Written informed consent was obtained from subjects or families of all participants. Hierarchical steatosis and steatohepatitis were independently diagnosed by two pathologists according to the scoring system of standard histological criteria established by the NASH Clinical Research Network[65]. Cases with NASH activity scores (NAS) of 1–3, and ballooning scores of 0 and no fibrosis, were classified as simple steatosis. Cases with NAS > 4 or NAS of 3–4 but with fibrosis were classified as NASH. Cases with NAS of 0 were classified as normal.

### Adeno-associated virus 8 construction and injection

The AAV8 delivery system was used to overexpress WT SENP1 and C599S mutant SENP1 in the livers of mice. AAV expressing WT SENP1 (pAAV8-CBh-EGFP-P2A-SENP1-3×FLAG-tWPA) or C599S SENP1 (pAAV8-CBh-EGFP-P2A-SENP1(C599S)−3×FLAG-tWPA) were generated by transfecting three plasmids (pAAV containing the SENP1 gene flanked by the AAV inverted terminal repeat sequences, pAAV8 trans-plasmid with the AAV rep and cap genes, and the pAAV helper plasmid) in HEK293T cells. The titers of the vector genome were measured via qPCR with vector-specific primers. Mice were injected via the tail vein with 100 µl of virus containing $2 \times 10^{11}$ vg of the AAV8 vector genomes. All AAVs were purchased from Taitool Biotech (Shanghai).

### Cell culture and generation of knockdown and reconstitution lines

MEFs (ATCC, CRL-2991), and HEK293T (ATCC, CRL3216) cells were cultured in DMEM (Gibco) with 10% (vol/vol) FBS (Gibco) and 100 units/ml penicillin/streptomycin (Invitrogen). All cells were cultured at 37 °C with 5% $CO_2$. SENP1 knockdown MEFs were generated with lentivirus transduction of SENP1 shRNA construct and selected with 2 µg/ml puromycin. SENP1 and PIAS1 double knockdown MEFs were generated on SENP1-KD MEFs with lentivirus transduction of PIAS1 shRNA construct and selected with 140 µg/ml hygromycin. RIPK1 WT or K550A mutant MEFs were generated by stably reconstituting *Ripk1*$^{-/-}$;*Senp1*$^{KD}$ MEFs with retrovirus transduction of RIPK1 WT or K550A mutant constructs and selected with 140 µg/ml hygromycin. *Senp1*$^{KD}$;SUMO3$^{HA}$ MEFs and *Senp1*$^{KD}$;*Pias1*$^{KD}$;SUMO3$^{HA}$ MEFs were generated on *Senp1*$^{KD}$ MEFs and *Senp1*$^{KD}$;*Pias1*$^{KD}$ MEFs, respectively, with lentivirus transduction of HA-tagged SUMO3 construct without selection.

### Primary hepatocyte isolation and culture

8-week-old male mice were used for isolation of primary hepatocytes. Mice were anesthetized with 90 mg/kg body weight of pentobarbital sodium first. Livers were fully digested via portal vein perfusion using Liver Perfusion Medium (17701-038; Life Technologies) and then Liver Digest Medium (17703-034; Life Technologies) at a rate of 2 ml/min for 5 min for each medium. After digestion, liver was excised, minced, and filtered through a 70 µm cell strainer (352350; Falcon). Primary hepatocytes were then separated via centrifugation at $50 \times g$ for 5 min and purified on 50% Percoll solution (P1644; Sigma). Primary hepatocytes were cultured in RPMI-1640 (Gibco) with 10% (vol/vol) FBS (Gibco) and 100 units/ml penicillin/streptomycin (Invitrogen) and were maintained in a humidified incubator under the same condition as the cell lines.

### Immunoblotting

Antibodies against the following proteins were used for western blot analysis: p-S166 RIPK1 (Biolynx, BX60008, 1:1000), RIPK1 (CST, 3493, 1:1000), PIAS1 (CST, 3550, 1:1000), A20 (CST, 5630, 1:1000), TNFR1 (CST, 13377, 1:1000), cleaved-caspase-3 (CST, 9661, 1:1000), p-p44/42 MAPK (CST, 4370, 1:1000), p-p38 (CST, 9216, 1:1000), p-JNK (CST, 4671, 1:1000), p-IκBα (CST, 2859, 1:1000), IκBα (CST, 4814, 1:1000), HA-tag (CST, 3724, 1:1000), RIPK3 (CST, 95702, 1:1000, 1:500 for IP), p-T231/S232 RIPK3 (CST, 91702, 1:1000), p-S345 MLKL (Abcam, ab196436, 1:1000), HOIP (Abcam, ab46322, 1:1000), SENP1 (Abcam, ab108981, 1:1000), FADD (Abcam, ab124812, 1:1000), FADD (Santa Cruz, SC-6036, 1:100 for IP), SENP1 (Santa Cruz, sc-271360, 1:1000), TRADD (Santa Cruz, sc-7868, 1:500), ERK1/2 (Proteintech, 67170-1-Ig, 1:1000), JNK (Proteintech, 24164-1-AP, 1:1000), p38 (Proteintech, 14064-1-AP, 1:1000), Caspase-3 (Proteintech, 19677-1-AP, 1:1000), SAE1 (Proteintech, 10229-1-AP, 1:1000), SHARPIN (Proteintech,14626-1-AP, 1:1000), β-Tubulin (TRANS, HC101-02, 1:10,000), SUMO3 (PTM-BIO, PTM-1109, 1:1000), Flag-tag (Sigma, F7425, 1:1000), Myc-tag (Sigma, C3956, 1:1000). Anti-K63 Ub and anti-M1 Ub were gifts from Dr. Vishva Dixit (Genentech, 1:1000 for IP). The signals were detected by Immobilon ECL Ultra Western HRP Substrate (Millipore). The membranes were reprobed after incubation in Restore Western Blot stripping buffer (21063, Thermo).

## Construction of plasmids and transfection

The annealed shRNA oligonucleotides were cloned into the pLKO.1 vector. To knockdown SENP1 in MEFs, the following shRNAs were used: sh*Senp1*−1 (5′-CGCAAAGACATTCAGACTCTA-3′); sh*Senp1*−2 (5′-GCCA-TATTTCCGAAAGCGAAT-3′).

To knockdown PIAS1 in MEFs, the following shRNAs were used: 5′-GCGTCCATCTTTGGCATCATA-3′. The K550A mutant of RIPK1 was created by PCR-based site-directed mutagenesis, and then cloned into the pMSCV vector. For protein expression, SENP1 (residues 415-644), USP2 (residues 271-618) and Ubc9 were cloned into NdeI/EcoRI sites in pET-28a plasmid for *E.coli* expression. All plasmids were confirmed by sequencing.

For plasmid transfection, transient transfections of HEK293T cells were performed using polyethylenimine (Polysciences) according to the manufacturers' instructions. For siRNA transfection, MEFs were transfected with 50 nM siRNA using Lipofectamin RNAiMax (Invitrogen) for 48 h following the manufacturer's instruction. The sense sequences of siRNAs used in this study were as follows: si*Senp1-1* (5′-GCAGUUCUGUGUAGCGAAATT-3′); si*Senp1-2* (5′-GCAGGAACAUGCA-GUACUUTT-3′). The knockdown efficiency was examined using western blot.

## Lentivirus and retrovirus production and infection

For lentiviral production, HEK293T cells were transfected with pLKO.1 vector carrying shRNA targeting SENP1 or PIAS1 with pMD2.G and psPAX plasmids for 48 h. For retroviral production, HEK293T cells were transfected with pMSCV vector carrying cDNA encoding RIPK1 (WT and K550A), or HA-SUMO3 with VSVG and GAG plasmids for 48 h. Harvested supernatant media from transfected HEK293T cells was filtered through a 0.45 μm filter. Filtered media containing lentivirus or retrovirus particles was used to infect target cells in the presence of polybrene (8 mg/ml). After 6 h of incubation with the lentivirus or retrovirus, the medium was replaced with fresh medium and selected with puromycin or hygromycin after 24 h of infection.

## Analysis of cytotoxicity and viability

For rates of cell death, MEFs were seeded day before at 2500 cells per well in a 384-well plate. The next day, cells were pretreated with the indicated compounds for 1 h. and then stimulated with TNFα in the presence of 5 mM SytoxGreen (Invitrogen). SytoxGreen intensity was measured at intervals of 30 min to 2 h using a SYNERGY H1 microplate reader (BioTek), with an excitation filter of 485 nm, emission filter of 520 nm. Data was collected by Gen5 software version 3.08.01 (BioTek). Percentage of cell death was calculated as (induced fluorescence−background fluorescence)/(max fluorescence−background fluorescence)×100. The maximal fluorescence is obtained by full permeabilization of the cells using Triton X-100 at a final concentration of 0.1%. The rates of cell viability were determined by using CellTiter-Glo Luminescent Cell Viability Assay (Promega) following the manufacturer's protocol and the results are expressed as percentages of luminescence intensity per well after deducting the background signal in blank well and compared to that of the viability in the non-treated wells.

## siRNA screen

siRNA screen was performed using 14 siRNA pools (GenePharma) targeting 14 rate-limiting enzymes/proteins that are involved in SUMOylation process. MEFs cultured in 384-well white plates were transfected with 50 nM siRNA by reverse transfection method using RNAiMax (Invitrogen) according to a protocol from the manufacturer. At 48 h after the transfection, the cells were treated with 100 nM 5Z-7-Oxozeaenol (5z7) and TNFα (1 ng/ml) and cultured for an additional 12 h to induce RDA. Viability was measured using luminescence-based ATP levels as a surrogate marker in surviving cells using CellTiterGlo ATP assay (Promega). Positive control (RIPK1) and negative control

(nontargeting control siRNA, GenePharma) were present in the same plate. The screen was performed in duplicate. Z-scores were calculated based on plate median (negative controls excluded) and median absolute deviation, with Z-score = (cell ATP value - median plate ATP value)/(plate median absolute deviation×1.4826)[66]. The screen hits were selected based on the median Z-score of the duplicate plates with cut-offs set at Z-score >2 or < −2 ($p < 0.05$).

## Immunoprecipitation

Cell lysates were prepared in the following lysis buffer: 50 mM Tris-HCl (pH 7.5), 150 mM NaCl, 1% NP40, 5% glycerol supplemented with phosphatase and protease inhibitor cocktail tablets (Bimake). Cells were lysed on ice for 30 min and centrifuged at 12,000 × *g* for 15 min at 4 °C. The cell lysates were incubated with indicated antibody overnight at 4 °C and immunocomplex was captured by protein A/G agarose (Invitrogen). After extensive washes, beads were boiled in SDS reducing sample buffer and eluted products were separated by SDS-PAGE which was transferred to PVDF membrane (Millipore) and analyzed with indicated antibodies.

For TNF-RSC (complex I) immunoprecipitations, MEFs were seeded in 15 cm dishes and treated as indicated with Flag-TNFα (100 ng/ml). To terminate treatment, media was removed and plates were washed two times in ice-cold PBS. Cells were lysed in 1 ml 0.5% NP-40 lysis buffer (50 mM Tris-HCl pH 7.5, 150 mM NaCl and 0.5% NP-40, 5% glycerol) supplemented with phosphatase and protease inhibitors and N-Ethylmaleimide (2.5 mg/ml, Sigma). Cell lysates were rotated at 4 °C for 30 min and then clarified at 4 °C at 12,000 × *g* for 15 min. Proteins were immunoprecipitated from cleared protein lysates with 20 μl of anti-Flag M2 beads (Sigma) with rotation overnight at 4 °C. 3× washes in 0.5% NP-40 buffer were performed, and samples were eluted by boiling in 50 μl 1× SDS loading buffer and analyzed by western blot.

For tandem immunoprecipitation of K63 and M1 ubiquitinated RIPK1 in TNF-RSC, MEFs were seeded in 15 cm dishes and treated as indicated with Flag-TNFα (100 ng/ml). TNF-RSC was immunoprecipitated as above and then eluted with 6 M urea. The resulting eluate was incubated with anti-K63 Ub antibody in NP-40 buffer containing 3 M urea or anti-M1 Ub antibody in NP-40 buffer containing 6 M urea overnight at 4 °C followed by incubation with protein A/G agarose resin at 4 °C for 4 h. The immune complexes were then eluted in 1× SDS loading buffer and analyzed by western blot.

For complex-II immunoprecipitations, MEFs were seeded in 10 cm dishes and treated as indicated. Cells were washed two times in ice-cold PBS before lysis in 1 ml 0.5% NP-40 lysis buffer. Cell lysates were rotated at 4 °C for 30 min then clarified at 4 °C at 12,000 × *g* for 15 min, and the supernatant was then incubated overnight at 4 °C with 2 μl anti-FADD (Santa Cruz, SC-6036, for complex IIa) or anti-RIPK3 antibodies (for necrosome). Then the immunocomplex was captured by protein A/G agarose (Invitrogen) for 4 h at 4 °C. The immune complexes were then eluted in 1 × SDS loading buffer and analyzed by western blot.

For tandem immunoprecipitation of SUMOylated RIPK1 in TNF-RSC, MEFs stably expressing HA-SUMO3 were seeded in 15 cm dishes and treated as indicated with Flag-TNFα (100 ng/ml). TNF-RSC was immunoprecipitated as above and then eluted with 3× Flag peptide at 25 °C for 1 h. The resulting eluate was incubated with 20 μl of anti-HA magnetic beads (Invitrogen) with rotation overnight at 4 °C. 3× washes in 0.5% NP-40 buffer were performed, and then samples were eluted by boiling in 50 μl 1× SDS loading buffer and analyzed by western blot.

## Ni-NTA pulldown assay

HEK293T cells were transfected with the indicated plasmids. Cells were washed two times in ice-cold PBS before lysis in 1 ml His lysis buffer (pH 8.0) containing 50 mM Tris, 150 mM NaCl, 8 M urea, 10% glycerol, 0.1% Triton X-100, 10 mM imidazole, 10 mM β-mercaptoethanol. Cells were clarified at 12,000 × *g* for 10 min, and the supernatant was then

incubated with 40 µl Ni-NTA agaroses (QIAGEN, 30210) for 4 h. Then the agaroses were washed with His wash buffer (pH 8.0) containing 50 mM Tris, 150 mM NaCl, 8 M urea, 10% glycerol, 0.1% Triton X-100, 20 mM imidazole, 10 mM β-mercaptoethanol, followed by boiling in 50 µl elution buffer (pH 8.0) containing 1 × SDS loading buffer, 1 M imidazole, 500 mM NaCl, 20 mM Tris and analyzed by western blot.

## Quantitative reverse-transcription PCR

Total RNA was isolated from E14.5 fetal livers and livers or hepatocytes of adult mice using FastPure Cell/Tissue Total RNA Isolation Kit (Vazyme). RNA concentration was measured using the Nanodrop spectrophotometer (Thermo scientific). cDNA was prepared with HiScript III RT SuperMix kit (Vazyme). cDNA (10 ng) of each sample was used for quantitative PCR with ChamQ Universal SYBR qPCR Master Mix (Vazyme). Quantitative reverse-transcription PCR of indicated genes was performed with QuantStudio 6 Flex Real-Time PCR System (Applied Biosystems). Data was collected by using QuantStudio 12K Flex software version 1.3 (Applied Biosystems). Data were analyzed according to the $\Delta CT$ method. The sequences of gene-specific primers used for PCR are shown below.

Mouse *Senp1* forward: CTGGGGAGGTGACCTTAGTGA, reverse: GTGATAATCTGGACGATAGGCTG; *Tnf* forward: CCCTCACACTCA-GATCATCTTCT, reverse: GCTACGACGTGGGCTACAG; *Il1a* forward: CGAAGACTACAGTTCTGCCATT, reverse: GACGTTTCAGAGGTTCT CAGAG; *Il1b* forward: GCAACTGTTCCTGAACTCAACT, reverse: ATCTTTTGGGGTCCGTCAACT; *Il6* forward: TAGTCCTTCCTACCCCA ATTTCC, reverse: TTGGTCCTTAGCCACTCCTTC; *Cxcl1* forward: CTGGGATTCACCTCAAGAACATC, reverse: CAGGGTCAAGGCAAGC CTC; *Ccl2* forward: TTAAAAACCTGGATCGGAACCAA, reverse: GCAT-TAGCTTCAGATTTACGGGT; *Ccl5* forward: GCTGCTTTGCCTAC CTCTCC, reverse: TCGAGTGACAAACACGACTGC; *Col3a1* forward: CTGTAACATGGAAACTGGGGAAA, reverse: CCATAGCTGAACTG AAAACCAC; *Acta2* forward: GTCCCAGACATCAGGGAGTAA, reverse: TCGGATACTTCAGCGTCAGGA; *Ccnd1* forward: GCGTACCCTGA-CACCAATCTC, reverse: CTCCTCTTCGCACTTCTGCTC; *Tgfb* forward: CTCCCGTGGCTTCTAGTGC, reverse: GCCTTAGTTTGGACAGGAT CTG; *Pdgfa* forward: GAGGAAGCCGAGATACCCC, reverse: TGCT GTGGATCTGACTTCGAG; *Timp1* forward: GCAACTCGGACCTGGTCA-TAA, reverse: CGGCCCGTGATGAGAAACT; *Pdgfra* forward: TCCATG CTAGACTCAGAAGTC, reverse: TCCCGGTGGACACAATTTTTC; *Pdgfb* forward: CATCCGCTCCTTTGATGATCTT, reverse: GTGCTCGGGTCAT GTTCAAGT; *Col1a1* forward: GCTCCTCTTAGGGGCCACT, reverse: CCACGTCTCACCATTGGGG; *Ripk1* forward: GAAGACAGACCTAGA-CAGCGG, reverse: CCAGTAGCTTCACCACTCGAC; *Alb* forward: TGCTTTTTCCAGGGGTGTGTT, reverse: TTACTTCCTGCACTAATTT GGCA; *Clec4f* forward: GAGGCCGAGCTGAACAGAG, reverse: TGTG AAGCCACCACAAAAGAG; *Tek* forward: GAGTCAGCTTGCTCCTT-TATGG, reverse: AGACACAAGAGGTAGGGAATTGA; *Epo* forward: AC TCTCCTTGCTACTGATTCCT, reverse: ATCGTGACATTTTTCTGCCTCC; *Epor* forward: GGGCTCCGAAGAACTTCTGTG, reverse: ATGACTTTC GTGACTCACCCT.

## Mass spectrometry and data analysis

For complex I mass spectrometry analysis, MEFs were treated with Flag-TNFα for indicated time. The binding proteins of TNFR1 in immunoprecipitation pulldown by anti-Flag beads were trypsin-digested on beads. The resulting peptides in duplicates were analyzed on a Thermo Scientific Orbitrap Fusion Tribrid mass spectrometer. The protein identification and quantification were done by MaxQuant 1.6. The tandem mass spectra were searched against the UniProt human protein database and a set of commonly observed contaminants. The precursor mass tolerance was set as 20 ppm, and the fragment mass tolerance was set as 0.5 Da. The cysteine carbami-domethylation was set as a static modification, and the methionine oxidation was set as a variable modification. The false discovery rate at

the peptide spectrum match level and protein level was controlled to be <1%. The unique peptides plus razor peptides were included for quantification. The summed peptide intensities were used for protein quantification.

For identification of SUMOylated proteins after TNFα stimulation, SUMOylated proteins isolated from *Senp1*^WT^; SUMO3^HA^ MEFs, and *Senp1*^KD^; SUMO3^HA^ MEFs stimulated by TNFα for 5 min were trypsin-digested on beads. The resulting peptides in triplicates were analyzed on a Thermo Scientific Orbitrap Fusion Tribrid mass spectrometer. The protein identification and quantification were done as above.

For mass spectrometry analysis of SUMOylation sites of RIPK1, Flag-tagged RIPK1 isolated from HEK293T cells expressing this construct was trypsin-digested on beads followed by immunoprecipitation. The resulting peptides were subjected to enrichment of diGly peptides using antibody against ubiquitin remnant motif (K-ε-GG) (PTM Biolabs Inc). The enriched diGly peptides were analyzed on the Q Exactive HF-X mass spectrometer (Thermo Scientific). The identification and quantification of diGly peptides were done by MaxQuant. The tandem mass spectra were searched against UniProt mouse protein database together with a set of commonly observed contaminants. The precursor mass tolerance was set as 20 ppm, and the fragment mass tolerance was set as 0.1 Da. The cysteine carba-midomethylation was set as a static modification, and the methionine oxidation as well as lysine with a diGly remnant were set as variable modifications. The FDR at peptide spectrum match level was controlled below 1%.

## RNA-seq and data analysis

Tissues were homogenized in a 1.5 ml tube containing 1 ml of Trizol Reagent (Invitrogen). Total RNA extraction was performed according to the manufacturer's instructions. RNA was resuspended in DEPC-treated RNase-free water and residual DNA contamination was removed by TURBO DNA-free kit according to manufacturer's instruction (Invitrogen). The quantity and quality of RNA were determined by Nanodrop (Thermo scientific) and agarose gel electrophoresis. 1 µg of total RNA was used for sequencing library preparation. PolyA-tailed RNAs were selected by VAHTS mRNA-seq V2 Library Prep Kit for Illumina (Vazyme) according to manufacturer's instruction. The library quality was examined by Bioanalyzer 2100 (Agilent). Quantification was performed by qRT-PCR with a standard library as reference. The libraries were pooled together in equimolar amounts to a final 2 nM concentration. The normalized libraries were denatured with 0.1 M NaOH. Pooled libraries were sequenced on the Illumina X-ten platforms with PE150 (Illumina). Data was collected by llumina X-ten platforms software version HCS 3.3.76 (Illumina).

Sequencing reads were mapped to the reference genome mm10 (GRCm38: GCA_000001635.8, GCF_000001635.26) from Gencode with hisat2 by default parameter. The read counts for each gene were calculated by featureCounts[67]. The count files were used as input to R package DESeq2 (v1.26.0)[68] for normalization and the differential expression genes. Wald test was performed. *P* value were corrected for multiple testing using the FDR method. Heatmap was plot by pheat-map R package. GO term analysis was performed by GSEA (v4.1.0). Enriched pathways were considered significant based on *q* values (<0.05; Fisher exact test).

## Protein expression and purification

SENP1 (residues 415-644), USP2 (residues 271-618), and Ubc9 were expressed as 6× His fusion protein in *E. coli* BL21 (DE3) with 0.1 mM IPTG overnight at 18 °C. Bacteria were centrifugation and pellets were lysed with sonication at 4 °C, and then purified by Ni-NTA affinity resin. All proteins were further purified by size exclusion chromatography in a buffer containing 50 mM Tris-HCl (pH 7.5), 300 mM NaCl, and 1 mM DTT.

## In vitro SUMOylation and deSUMOylation assay

Flag-RIPK1 was isolated from HEK293T expressing this construct using anti-Flag M2 beads and washed with IP buffer three times followed by incubation with 2 μg recombinant catalytic domain (CD) of His-USP2 in deubiquitination buffer [50 mM Tris-HCl (pH 8.0), 50 mM NaCl, and 5 mM DTT] at 37 °C for 1 h to remove the ubiquitination smears. The resulting mixture was then incubated in a reaction containing 50 nM recombinant human SUMO activating enzyme E1 (SAE1/UBA2, R&D Systems), 2 μg E2 enzyme (His-Ubc9), 10 μM recombinant SUMO3 (R&D Systems) and 10 mM ATP with SUMOylation buffer (50 mM Tris-HCl pH 7.5, 5 mM $MgCl_2$) at 37 °C for 1 h. The samples were then analyzed by immunoblotting with RIPK1. For in vitro deSUMOylation assay, SUMOylated RIPK1 was prepared as above, and then incubated with 0.5 μM recombinant catalytic domain (CD) of His-SENP1 in deSUMOylation buffer containing 50 mM Tris-HCl (pH 7.5), 2 mM DTT at 37 °C for 4 h. The samples were then analyzed by immunoblotting with RIPK1.

## TUNEL tissue staining

Collected mouse tissue was fixed in 4% PFA and processed for paraffin embedding. TUNEL assay was used to detect dead cells with DNA fragmentation using In Situ Cell Death Detection Kit, POD (Roche) by following manufacture's protocol.

## Biochemical serum analysis

Alanine aminotransferase (ALT) and aspartate transaminase (AST) were measured in blood serum using Cobas C111 biochemical analyzer (Roche, Mannheim, Germany) according to the manufacturer's instructions.

## Hepatic and intracellular lipid analyses

For hepatic lipid content, TG and TC content was examined using commercial kits (No. 290-63701 for TG, and No. 294-65801 for TC, Wako, Tokyo, Japan) according to the manufacturer's instructions. Intracellular TG levels were measured using a commercially available TG Colorimetric Assay Kit (Cayman, Ann Arbor, MI, USA) according to the manufacturer's protocol.

## Histology and immunochemistry

Tissues were harvested from mice with different genotypes and fixed with 4% paraformaldehyde. Fixed tissues were embedded with paraffin. Sections were dewaxed and then antigen retrieval was performed with 0.01 M sodium citrate. The sections were firstly blocked with 3% $H_2O_2$ and next with 5% goat serum and incubated with primary antibodies at 4 °C overnight and washed in PBST before incubating with secondary antibodies at RT for 2 h. For immunostaining, fluorescent images were collected by the Leica TSC SP8 confocal microscopy system (Leica Application Suite X software version 1.8.1.13759) using a 20× or 40× objective. For immunochemistry, the signals were detected using SignalStain DAB Substrate Kit (CST, 8059). The following antibodies were used: p-S166 RIPK1 (Biolynx, BX60008, 1:500), CD45 (Servicebio, GB11066, 1:500), EPO (Servicebio, GB11323, 1:900), CC3 (CST, 9661, 1:400). Secondary antibodies: Alexa Fluor 488 goat anti-rabbit IgG (Invitrogen, A11034, 1:2000); Alexa Fluor 568 goat anti-rabbit IgG (Invitrogen, A11011, 1:2000).

## Oil red O staining

Oil red O (ORO) staining was performed to detect neutral lipids and lipid droplet morphology with slight modifications. Tissues were harvested from mice with different genotypes and fixed with 4% paraformaldehyde overnight at 4 °C, rinsed in PBS, and incubated overnight at 4 °C in 30% sucrose before freezing in a 2:1 mixture of 30% sucrose and Tissue-Tek optimal cutting temperature compound (OCT). The frozen liver sections that were prepared using OCT compound were immersed in freshly prepared 1% Oil red O working solution for 10 min,

counterstained with hematoxylin, and then rinsed under running tap water for 30 min. Photomicrographs were captured under a microscope.

## Masson's trichrome staining

Masson's trichrome stain was used to determine the levels of collagen deposition. Tissues were harvested from mice with different genotypes and fixed with 4% paraformaldehyde. Fixed tissues were embedded with paraffin, sliced into 5-μm sections, and stained with Masson's trichrome at room temperature for 2 h. All hepatic sections were examined and images were captured via light microscopy.

## Statistics and reproducibility

All cell death data are presented as mean ± s.d. of one representative experiment. Each experiment was repeated at least three times independently with similar results. Mouse data are presented as mean ± s.e.m. of the indicated $n$ values. All immunoblots were repeated at least three times independently with similar results. Quantifications of immunoblots were performed using ImageJ 1.52a, and the densitometry data were adjusted to loading control and normalized to control treatment. Statistical analyses were performed with GraphPad Prism 8.0 (v8.4.1). Normality of the samples was checked by using the Shapiro-Wilk test before statistical analysis. For normal distribution, either unpaired two-tailed Student's t-test for comparison between two groups, or one-way ANOVA with post hoc Dunnett's tests for comparisons among multiple groups with a single control was applied, while for non-normal distribution, a non-parametric statistical analysis was performed using Mann-Whitney test for two groups, or Kruskal-Wallis test followed by Dunnett's test for multiple comparisons. Multiple linear regression analysis was performed to assay the correlation of hepatic SENP1 expression level with variations in NAFLD and NASH. Differences were considered statistically significant if $P < 0.05$(*); $P < 0.01$ (**); $P < 0.001$(***).

## Reporting summary

Further information on research design is available in the Nature Portfolio Reporting Summary linked to this article.

## Data availability

The RNA-seq data generated in this study have been deposited at Gene Expression Omnibus (GEO) database under accession code GSE193509. The mass spectrometry proteomics data have been deposited in the ProteomeXchange Consortium database under accession code PXD037648. These data are available with no restriction. The clinical and histological characteristics of the human samples are not publicly available due to data privacy, but that anonymized data can be obtained from the corresponding author D.X. (xudaichao@sioc.ac.cn) and J.G. (gjynyd@126.com) immediately upon reasonable request with no restriction. All other data generated or analyzed during this study are included in this published article and its supplementary information files. Source data are provided with this paper.

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

## Acknowledgements
We thank Drs. Jinke Cheng (Shanghai Jiao Tong University College of Basic Medical Sciences, China) for *Senp1*[+/−] and *Senp1*[flox/flox] mice, and Vishva Dixit (Genentech) for anti-M1 and K63 ubiquitin antibodies. This work was supported in part by grants from the Strategic Priority Research Program of the Chinese Academy of Sciences (XDB39030600), the National Key R&D Program of China (2022ZD0213200), the National Natural Science Foundation of China (32070737 and 92049303), the Shanghai Science and Technology Development Funds (20JC1411600, 20QA1411500 and 22JC1410400), the Shanghai Key Laboratory of Aging Studies (19DZ2260400) and Shanghai Municipal Science and Technology Major Project (2019SHZDZX02). The work of J.G. and N.L. was supported in part by the National Natural Science Foundation of China (82130020 and 82072645 to J.G., 91849109 to N.L.).

## Author contributions
D.X. conceived, designed, and directed this project. L.Y. designed and conducted majority of the experiments. T.Z. conducted key experiments during revision. J.G., Y.C.Z., and K.W. assisted with experiments on human liver samples. Z.C. assisted with RNA-seq data analysis. Y.Y. assisted with cloning experiments. Q.S. assisted with animal experiments. B.S. and M.Z. conducted the mass spectrometry analysis. N.L. and Y.D.Z. assisted with the RNA-seq experiment. D.X., J.G., and L.Y. wrote the manuscript.

## Competing interests
The authors declare no competing interests.
