## [Peer Review File · Nature Communications]

SENP1 prevents steatohepatitis by suppressing RIPK1-driven apoptosis and inflammationREVIEWER COMMENTS

Reviewer #1 (Remarks to the Author):

The manuscript by Yan et al. describes how SENP1 suppresses RIPK1-driven apoptosis and inflammation in NASH. Initially the authors show that SENP1 expression levels in liver is associated with the severity of NAFLD. In addition, they provide evidence that the hepatocyte-specific SENP-1 KO mice develop spontaneous liver damage, steatosis, inflammation, and fibrosis in a RIPK1 kinase-dependent manner. These *in vivo* studies are followed by an elegant set of *in vitro* experiments in MEFs where the authors show that RIPK1 is SUMOylated at K550 and SUMOylation of RIPK1 promotes its activation. Finally, the authors describe that in the *Senp1^{-/-};Ripk1D138N/D183N* mice, RIPK1 inactivation inhibits NAFLD induced by a high fat diet in SENP-1 deficient mice. These results are significant in the field and present novel data relative to the current literature, a field that has been gaining research interest in the last couple of years.

Even though this study reports a novelty mechanistic regulation of RIPK1 through SENP1 in NASH, data analysis, interpretation, and conclusions together with the methodology are sound, some further evidence is needed to support the conclusions and claims. Mostly, SUMOylation of RIPK1 is not shown *in vivo*. This should be explored both in human samples of NASH patients and in the NAFLD mouse models. For this end, both mass spectrometry studies or immunoprecipitation assays could be carried out. Moreover, it would be interesting to perform some of the *in vitro* studies not only in MEFs but also in primary hepatocytes or human hepatocytes cell lines upon stimuli with different steatosis inducers, such as for example FFA, described to induce TNF- α expression (Feldstein et al., *Hepatology*, 2004), and in these *in vitro* models assess SENP1 levels, RIPK1 SUMOylation and phosphorylation. This would further validate the mechanism proposed in here for RIPK1 regulation in the context of NASH.

Also, the authors should show the correlation between SENP1 levels and the fibrosis stage as the NAS score does not include the fibrosis stage. Especially considering the previous reported papers on the role of *senp2* in hepatic stellate cell apoptosis and reversion of liver fibrosis (Toxicol Lett 2018). In line with this, in extended figure 1a the authors show that whole liver extracts of *Senp1f/f, Alb-Cre* present null values of SENP1. However, silencing is exclusive of hepatocytes and according to protein atlas, SENP1 is present in other hepatic cell types, such as Kupffer cells and endothelial cells. Have the authors explored the SENP1 levels during NAFLD in other types of hepatic cells?

In line 94-95, the authors say that given an intimate association of SENP1 with NASH features. However, literature data on SENP1 and NASH is scarce with the two references of the relevance of SENP1 in a rat model of liver fibrosis (Wu et al. 2018) and in adipocytes (Shao et al. 2015 *Nature Communications*). In addition, the authors show that the *alb cre senp1^{-/-}* mice present NASH. Therefore, it would be interesting to assess the levels of SENP1 in different mouse models of NAFLD, such as the HFD mice used in this study.

In line 155 the authors say that the inhibition of RIPK1 reduces liver fibrosis. However, this is not shown. What the author show is that inhibition of RIPK1 is associated with lower fibrosis which could be related to prevention of fibrosis because of low hepatocellular death and inflammation. If the authors want to show fibrosis reversion another experimental design should be carried out. Meanwhile, this section should be rewritten.

In Figure 2b the authors show that 3-mo mice show elevation of transcriptomics related to collagen deposition. However, no histology is shown on collagen at this age, and this should be added to the manuscript to understand the temporal pathogenesis of the disease.

In line 253-254. The authors state that SENP1 KD also increased RIPK1 ubiquitination (Fig. 4a), which cannot be blocked by the addition of Nec-1s (Fig. 4b). How do authors quantify the ubiquitination in

the presented WB?

Minor comments include:

In Fig. 1c the authors should maintain the same color scheme to confirm that ALT and AST levels are also related to the disease stage.

Line 130. SNEP1 instead of SENP1

Line 104. The title should state hepatocytes specific instead of liver-specific.

Line 129. the title should be reordered to highlight the stages involved in NAFLD progression, first steatosis, inflammation, and then fibrosis.

Reviewer #2 (Remarks to the Author):

The manuscript by Yan et al demonstrated that SENP1 regulated hepatic apoptosis and inflammation through governing SUMOylation-dependent RIPK1 activation. Hepatic deficiency of SENP1 activated RIPK1 and resulted in severe fibrosis and inflammation in the liver; while inhibition of RIPK1 totally reversed the phenomena. However, considering NASH is developed from steatosis, the phenotype in SENP1 hepatic KO mice was not a typical NASH, which might primarily ascribe to the apoptosis and inflammatory response. One of the important finds by the author is the elucidation that RIPK1 sumoylation in TNF-RSC, regulated by SENP1 and PIAS1, promoted its ubiquitylation and activation, thereby affecting apoptosis in hepatocytes. Overall, most of the experiments are well-designed and interpretations for the most part are supported by the evidence. Some suggestions below will hopefully help strengthen the paper.

1. The author should think over the title of manuscript regarding "NASH". NASH is notably characterized by steatosis, and then hepatocyte injury and inflammation. But it seems as if inflammatory response and apoptosis in SENP1 hepatic KO mice happened before hepatic steatosis. So it might be not typical NASH.
2. The author demonstrated the negative correlation between hepatic SENP1 expression and severity of NAFLD in human specimen. SENP1 expression in mice NAFLD or NASH model should be detected to strengthen the results.
3. Senp1f/f;Alb-Cre mice are liver specific, while Ripk1D138N/D138N mice are whole body RIPK1 inactivation. To exclude the possibility that inactive RIPK1 in other cell (in addition to hepatocyte) rescued the phenotype Senp1f/f;Alb-Cre mice, the author should isolated hepatocyte from Senp1f/f;Alb-Cre; Ripk1D138N/D138N mice and detected steatosis, inflammatory and apoptosis; or using liver-specific AAV to inhibit hepatic Ripk1 activity and conducted similar measurement.
4. The author provided solid data to prove that sumoylation of RIPK1 in TNF-RSC promoted its ubiquitylation and activation, and also successfully identified the sumoylated residue in RIPK1. Given sumoylation of RIPK1 in TNF-RSC reached its peak in 5min upon TNF α treatment in WT cells (fig. 5d), the author should explain why SENP1 was recruited at this time course (fig. 5e). Perhaps, PIAS1 was recruited first to activate RIPK1, and then SENP1 involved in to limit signal overamplification? The author should test this sequence by detecting whether PIAS1 was recruited before 5 min, and then SENP1.
5. In according to the current data, PIAS1 was required for RIPK1 activation. However, it is well-established that PIAS1 negatively regulated inflammation. Please make discussion about this difference.
6. To test the importance of desumoylation activity of SENP1 in the current model, SENP1 and its catalytically inactive C603S mutant (SENP1-C603S) should be replenished in the Senp1f/f;Alb-Cre mice to test RIPK1 activation, and steatosis, inflammatory and apoptosis phenotype in the liver.
7. The writing of the manuscript should be improved, with some mistakes need to be revised. such as:

line 130, hepatic "SNEP1" deletion. Experimental results should be described in order. For example, figure 1e was described after Figure 1f and 1g by the author.

8. Sample size should be expanded, especially for the in vivo measurements in fig. 3 and fig. 7, in which only three mice were included in some groups.

Reviewer #3 (Remarks to the Author):

In this manuscript Yan et al decipher the role and the regulation of RIPK1-SUMOylation which was hitherto unknown and show the protective role of SENP1 in hepatocyte during liver inflammation in NASH. They performed several in vitro and in vivo studies to understand the regulation of RIPK1 SUMOylation by SENP1 and PIAS1 and its incidence on TNF-induced cell death. They found that de-SUMOylation of RIPK1 by SENP1 inhibits its activation and TNFa-induced apoptosis. Conversely, in SENP1 KO PIAS1 promotes RIPK1 SUMOylation and sensitizes to TNF induced cell death. Overall, data are convincing and offer a persuasive mechanism on the unknown role of RIPK1 SUMOylation. However, some conclusions are overstated and additional experiments are necessary to improve the manuscript.

The authors demonstrate the protective role of SENP1 in NASH. Although the reduction of SENP1 during disease progression is convincing, it is unlikely that the role of SENP1 is restricted to metabolic liver disease. SENP1 inhibits RIPK1 apoptosis induced by TNFa stimulation. As TNFa is a very common cytokine in the inflamed liver (no matter the etiology), it would be expected that SENP1 would protect hepatocyte from cell death in other type of liver disease. To confirm this hypothesis, it would be interesting to see if LPS triggers liver injury in mice as well. Statements in the manuscript should be modified accordingly.

The authors claim that SENP1 promotes lipid deposition, however the oil red staining (only marker used in this study) does not seem to show specific staining for lipid droplets but conversely the all tissue seems to be red. Additional experiments such as liver triglyceride and liver cholesterol measurements are needed for this statement.

All the in vitro studies were done on Mouse Embryonic Fibroblasts whereas the in vivo analysis focuses on hepatocytes. In order to link both sections, it would be interesting to see that SENP1 KO primary hepatocytes are equally sensitive to TNFa in vitro and protected by RIPK1 kinase inhibition.

Extended 2E, the authors claim that SENP1 does not affect NF- κ B activation. However, knockdown with shSenp1-2 (the most potent shRNA regarding SENP1 expression) seems to reduce p-I κ B α (S32) and p-JNK. These results should be confirmed and quantified.

The authors show that SENP1 KO cells are sensitized to TNF-induced apoptosis or necroptosis, when caspases are inhibited. As apoptosis and necroptosis are two cell death mediators observed in NASH, what type of cell death is induced in Senp1f/f;Alb-Cre mice?

Minor comments:

- Line 130: change SNEP1 to SENP1
- Method for preparing frozen blocks is missing
- Statistical analysis: Does the distribution of the samples was checked before statistical analysis? Test should be different if the samples are normally distributed or not.
- Figure 1e – DAPI can be hardly seen – provide a higher exposure picture. In addition RIPK1 (S166) staining seems to stain non-parenchymal cells. Did p-RIPK1 was also observed in hepatocytes?

Point-to-Point Responses to the Reviewers' Critiques (#NCOMMS-22-10617-T)

Reviewer #1 (Remarks to the Author):

The manuscript by Yan et al. describes how SENP1 suppresses RIPK1-driven apoptosis and inflammation in NASH. Initially the authors show that SENP1 expression levels in liver is associated with the severity of NAFLD. In addition, they provide evidence that the hepatocyte-specific SENP1 KO mice develop spontaneous liver damage, steatosis, inflammation, and fibrosis in a RIPK1 kinase-dependent manner. These in vivo studies are followed by an elegant set of in vitro experiments in MEFs where the authors show that RIPK1 is SUMOylated at K550 and SUMOylation of RIPK1 promotes its activation. Finally, the authors describe that in the *Senp1*^{-/-}; *Ripk1*^{D138N/D183N} mice, RIPK1 inactivation inhibits NAFLD induced by a high fat diet in SENP1 deficient mice. These results are significant in the field and present novel data relative to the current literature, a field that has been gaining research interest in the last couple of years. Even though this study reports a novelty mechanistic regulation of RIPK1 through SENP1 in NASH, data analysis, interpretation, and conclusions together with the methodology are sound, some further evidence is needed to support the conclusions and claims.

Reply: We thank this reviewer for strong support and helpful suggestions! We have conducted additional experiments to address the questions raised by this reviewer. The new data further strengthened the conclusions!

Mostly, SUMOylation of RIPK1 is not shown in vivo. This should be explored both in human samples of NASH patients and in the NAFLD mouse models. For this end, both mass spectrometry studies or immunoprecipitation assays could be carried out.

Reply: To address this question, we performed IP assay to enrich RIPK1, and assessed the SUMOylation levels of RIPK1 by immunoblotting SUMO2/3 in human samples of NASH patients and in the NAFLD mouse models. We found that RIPK1 SUMOylation is substantially increased in the livers of individuals with NASH than that in the nonsteatotic controls (**Fig. R1a** or revised **Supplementary Fig. 6h**). We observed reduced SENP1 expression in the livers of mice with established NAFLD (**Fig. R1b** or revised **Supplementary Fig. 1a**), consistently, RIPK1 SUMOylation is also increased in the livers of mice with NAFLD than that in control mice (**Fig. R1c** or revised **Supplementary Fig. 6i**). These results suggest that RIPK1 is also SUMOylated in human samples of NASH patients and in the NAFLD mouse models that show reduced SENP1 expression. We thank the reviewer for this suggestion, which strengthened the paper.

Fig. R1. (a) Representative western blot analysis of RIPK1 SUMOylation in the livers of individuals with non-steatosis (n = 3) or NASH (n = 3). (b) Representative western blot analysis of SENP1 protein levels in the livers of mice fed a HFD for the indicated number of weeks (W). n = 4 mice per group. (c) Representative western blot analysis of RIPK1 SUMOylation in the livers of mice fed a HFD for the indicated number of weeks (W). n = 3 mice per group.

Moreover, it would be interesting to perform some of the *in vitro* studies not only in MEFs but also in primary hepatocytes or human hepatocytes cell lines upon stimuli with different steatosis inducers, such as for example FFA, described to induce TNF- α expression (Feldstein et al., *Hepatology*, 2004), and in these *in vitro* models assess SENP1 levels, RIPK1 SUMOylation and phosphorylation. This would further validate the mechanism proposed in here for RIPK1 regulation in the context of NASH.

Reply: To address this question, we stimulated the cultured primary hepatocytes with 0.4 mM palmitic acid (PA), as it is known that this concentration mimics some aspects of fatty liver disease *in vivo*. However, we did not observe remarkable reduction in SENP1 protein levels in primary hepatocyte treated with PA up to 72 hours (**Fig. R2a**), suggesting that the reduction of SENP1 in NASH *in vivo* may not be caused by PA per se. We further stimulated cultured primary hepatocytes derived from 2-m-o *Senp1^{fl/fl}* (SENP1-WT) and *Senp1^{fl/fl}, Alb-Cre* (SENP1-KO) mice with PA for 24 hours and assessed RIPK1 SUMOylation levels. We found that deletion of SENP1 in hepatocytes substantially increased RIPK1 SUMOylation levels as compared to WT hepatocytes after treatment of PA (**Fig. R2b** or revised **Supplementary Fig. 9f**). We also observed substantial cell death in SENP1-KO hepatocytes treated with PA, which is not presented in SENP1-WT hepatocytes (**Fig. R2c** or revised **Supplementary Fig. 9g**). Consistently, RIPK1 activation as determined by its S166 phosphorylation, and apoptosis as shown by cleaved-caspase-3 (CC3) are increased in SENP1-KO hepatocytes treated with PA (**Fig. R2d** or revised **Supplementary Fig. 9h**). In addition, the apoptosis of SENP1-KO hepatocytes induced by PA was fully protected by RIPK1-D138N mutation (**Fig. R2c, d** or revised **Supplementary Fig. 9g, h**). These results suggest that SENP1 deficiency in hepatocytes promotes RIPK1 SUMOylation, activation and apoptosis in response to PA *in vitro*. We thank the reviewer for this excellent suggestion, which strengthened the paper.

Fig. R2. (a) Representative western blot analysis of SENP1 expression in the primary hepatocytes treated with palmitic acid (PA, 0.4 mM) for indicated time. (b) Representative western blot analysis of RIPK1 SUMOylation levels in primary hepatocytes derived from 2-m-o *Senp1^{fl/fl}* and *Senp1^{fl/fl};Alb-Cre* mice treated with 0.4 mM PA for 24 hours. (c, d) SENP1 deficiency sensitized primary hepatocytes to cell death induced by PA. Primary hepatocytes derived from 2-m-o *Senp1^{fl/fl}*, *Senp1^{fl/fl};Alb-Cre*, *Senp1^{fl/fl};Alb-Cre;Ripk1^{D138N/D138N}* mice were treated with 0.4 mM PA for 24 hours. Cell death was measured by SytoxGreen positivity assay, data are represented as mean \pm s.d. of $n = 5$ independent samples of one representative experiment out of three independent experiments, One-way ANOVA, post hoc Dunnett's test (c). The levels of p-S166 RIPK1 and CC3 were determined by immunoblotting (d).

Also, the authors should show the correlation between SENP1 levels and the fibrosis stage as the NAS score does not include the fibrosis stage. Especially considering the previous reported papers on the role of senp2 in hepatic stellate cell apoptosis and reversion of liver fibrosis (Toxicol Lett 2018).

Reply: The NAS score in this manuscript is determined according to the scoring system of standard histological criteria established by the NASH Clinical Research Network (Kleiner, et al. *Hepatology*, PMID: 15915461), which does include the fibrosis stage. Cases with NAS of 1-3, and ballooning scores of 0 and no fibrosis, were classified as simple steatosis. Cases with NAS > 4 or NAS of 3-4 but with fibrosis were classified as NASH. Cases with NAS of 0 were classified as normal. The description on NAS score is included in Method section. The clinical and histological characteristics of the samples are provided in Source data file, which shows the fibrosis stage of each sample.

In line with this, in extended figure 1a the authors show that whole liver extracts of *Senp1^{fl/fl}; Alb-Cre* present null values of SENP1. However, silencing is exclusive of hepatocytes and according to protein atlas, SENP1 is present in other hepatic cell types, such as Kupffer cells and endothelial cells. Have the authors explored the SENP1 levels during NAFLD in other types of hepatic cells?

Reply: Because hepatocytes make up roughly 90% of the liver's cells, high knockout efficiency of a gene in hepatocytes may lead to almost null values of the protein when presented in western blot. This can be found in many cases (Verboom, et al. *Cell Rep*, PMID: 32075762, Fig. 2A; Van, et al. *JCI*, PMID: 28628031, Fig. 2A; Kondylis, et al. *Cancer Cell*, PMID:26555174, Fig. 1A, 3A, 7A, 8A). To further address this question, we isolated hepatocytes, Kupffer cells and endothelial cells from mouse liver and assessed the expression levels of SENP1 in these cells. We found that SENP1 level is higher in hepatocytes than that in Kupffer cells and endothelial cells (Fig. R3a or revised Supplementary Fig. 1b). Consistently, the expression of SENP1 almost disappeared in *Senp1^{fl/fl};Alb-Cre* hepatocytes (Fig. R3b or revised Supplementary Fig. 1d). To explore how SENP1 might

change in Kupffer cells and endothelial cells during NAFLD, we isolated hepatocytes, Kupffer cells and endothelial cells from livers of control mice and HFD-fed mice, respectively (**Fig. R3a**), and detected the expression levels of SENP1. We found that SENP1 is reduced in hepatocytes from NAFLD mice when compared to that of control mice (**Fig. R3a**). However, SENP1 was not reduced in Kupffer cells and endothelial cells from mice with NAFLD (**Fig. R3a**). Thus, SENP1 is specifically reduced in hepatocytes during NAFLD.

Fig. R3. (a) Western blot analysis of SENP1 expression in isolated hepatocytes, Kupffer cells and endothelial cells from livers of control mice and HFD-fed (24 weeks) mice. ND (n = 2), HFD (n = 2). The purity of isolated each type of cells was determined by measuring the hepatocyte-specific gene *Alb*, Kupffer cell-specific gene *Clec4f*, and endothelia cell-specific gene *Tek*. n = 6 mice per group. **(b)** Representative western blot analysis of SENP1 protein levels in primary hepatocytes isolated from livers with indicated genotypes. n = 3 mice per group.

In line 94-95, the authors say that given an intimate association of SENP1 with NASH features. However, literature data on SENP1 and NASH is scarce with the two references of the relevance of SENP1 in a rat model of liver fibrosis (Wu et al. 2018) and in adipocytes (Shao et al. 2015 Nature Communications).

Reply: We have modified this sentence as “Given an intimate association of SENP1 with inflammation and fibrosis, as well as the apparent functional role of SUMOylation in NASH, we hypothesized that SENP1 has a pivotal role in NASH pathogenesis as well.”

In addition, the authors show that the alb-cre senp1^{-/-} mice present NASH. Therefore, it would be interesting to assess the levels of SENP1 in different mouse models of NAFLD, such as the HFD mice used in this study.

Reply: To address this question, we assessed the levels of SENP1 in hepatocytes from mice with NAFLD, and found that SENP1 is reduced in hepatocytes from NAFLD mice when compared to that of control mice (**Fig. R3a** or revised **Supplementary Fig. 1b**). We also observed reduced SENP1 expression in whole liver extracts of mice fed a high-fat diet (HFD) (**Fig. R1b** or revised **Supplementary Fig. 1a**), which is consistent with the observation of SENP1 expression in human samples of NASH patients.

In line 155 the authors say that the inhibition of RIPK1 reduces liver fibrosis. However, this is not shown. What the author show is that inhibition of RIPK1 is associated with lower fibrosis which could be related to prevention of fibrosis because of low hepatocellular death and inflammation. If the authors want to show fibrosis reversion another experimental design should be carried out. Meanwhile, this section should be rewritten.

Reply: We agree with the reviewer that the prevention of fibrosis by RIPK1 inhibition is due to reduced hepatocellular death and inflammation. RIPK1 kinase-dependent inflammation and hepatocyte apoptosis has been shown to promote liver fibrosis, inhibition of RIPK1 lead to reduced fibrosis due to prevention of inflammation and hepatocyte apoptosis (Tan, et al. *PNAS*, PMID: 32513687; Kondylis, et al. *Cancer Cell*, PMID: 26555174). We next explored how RIPK1 kinase is involved in promoting *Senp1^{fl/fl}, Alb-Cre*-induced liver fibrosis. Because hepatocellular death and inflammation is known to activate hepatic stellate cells (HSC), which then release extracellular matrix proteins, including collagen, to promote the formation of fibrotic scars (Friedman, *Gastroenterology*, PMID: 18471545; Bataller, et al. *JCI*, PMID: 15690074), we next analyzed the expression of Desmin, the marker of HSC, in the livers of mice. As shown in **Fig. R4** (revised **Fig. 2h**), the numbers of HSC were increased in livers of 8-m-o *Senp1^{fl/fl}, Alb-Cre* mice compared to that of *Senp1^{fl/fl}* mice, which was reduced by RIPK1-D138N mutation. We have also rewritten this section. We thank the reviewer for this excellent suggestion.

Fig. R4. Immunohistochemistry of HSC marker Desmin in the livers of indicated genotypes. The quantification is shown at right (n = 5 mice for each genotype). Each dot represents an individual mouse. Mean \pm s.e.m. One-way ANOVA, post hoc Dunnett's test.

In Figure 2b the authors show that 3-m-o mice show elevation of transcriptomics related to collagen deposition. However, no histology is shown on collagen at this age, and this should be added to the manuscript to understand the temporal pathogenesis of the disease.

Reply: To address this question, we performed Masson's trichrome staining to evaluate the liver fibrosis at 3-month-old. Consistent with the induction of mRNA levels of fibrillary collagen in livers of 3-m-o *Senp1^{fl/fl}, Alb-Cre* mice, the fibrillary collagen deposition is also increased in the livers of 3-m-o *Senp1^{fl/fl}, Alb-Cre* mice (**Fig. R5** or revised **Supplementary Fig. 1o**).

Fig. R5. Representative images of Masson's trichrome stained (MTS) liver sections from 3-month-old mice of indicated genotypes. The quantification is shown at right (n = 5 mice for each genotype). Each dot represents an individual mouse. Mean \pm s.e.m. Unpaired two-tailed t-test.

In line 253-254. The authors state that SENP1 KD also increased RIPK1 ubiquitination (Fig. 4a), which cannot

be blocked by the addition of Nec-1s (Fig. 4b). How do authors quantify the ubiquitination in the presented WB?

Reply: Quantification of conjugated ubiquitin can be achieved by using the quantitative analysis (densitometric) of the smears (Chadchankar, et al. *PLoS One*, PMID:31703099). The quantification of conjugated ubiquitin of Fig. 4b is shown below (**Fig. R6**), which suggests that the addition of Nec-1s does not affect the ubiquitination levels of RIPK1 in TNF-RSC. This is also consistent with previous reports that inhibition of RIPK1 kinase does affect its ubiquitination levels in TNF-RSC (Newton, et al. *Nature*, PMID:27819682, Fig. 2c; Xu, et al. *Cell*, PMID: 30146158, Fig. 3E).

Fig. R6. Quantification of conjugated ubiquitin on RIPK1 from Fig. 4b. Each dot represents an independent experiment. Mean \pm s.e.m..

Minor comments include:

In Fig. 1c the authors should maintain the same color scheme to confirm that ALT and AST levels are also related to the disease stage.

Reply: We have used the same color scheme in Fig. 1c (**Fig. R7** or revised **Fig. 1c**).

Fig. R7. Pearson comparison analyses of the correlation between SENP1 protein level (normalized to Tubulin level) and serum ALT and AST concentrations.

Line 130. SNEP1 instead of SENP1

Reply: We have corrected this typo.

Line 104. The title should state hepatocytes specific instead of liver-specific.

Reply: We have changed “liver-specific” to “hepatocyte-specific” in this title.

Line 129. the title should be reordered to highlight the stages involved in NAFLD progression, first steatosis, inflammation, and then fibrosis.

Reply: We have reordered the title as well as the text in this part as the reviewer suggested.

Reviewer #2 (Remarks to the Author):

The manuscript by Yan et al demonstrated that SENP1 regulated hepatic apoptosis and inflammation through governing SUMOylation-dependent RIPK1 activation. Hepatic deficiency of SENP1 activated RIPK1 and resulted in severe fibrosis and inflammation in the liver; while inhibition of RIPK1 totally reversed the phenomena. However, considering NASH is developed from steatosis, the phenotype in SENP1 hepatic KO mice was not a typical NASH, which might primarily ascribe to the apoptosis and inflammatory response. One of the important finds by the author is the elucidation that RIPK1 sumoylation in TNF-RSC, regulated by SENP1 and PIAS1, promoted its ubiquitylation and activation, thereby affecting apoptosis in hepatocytes. Overall, most of the experiments are well-designed and interpretations for the most part are supported by the evidence.

Reply: We thank this reviewer for the support! We have worked hard to address all of the comments made by this reviewer.

Some suggestions below will hopefully help strengthen the paper.

1. The author should think over the title of manuscript regarding "NASH". NASH is notably characterized by steatosis, and then hepatocyte injury and inflammation. But it seems as if inflammatory response and apoptosis in SENP1 hepatic KO mice happened before hepatic steatosis. So, it might be not typical NASH.

Reply: We agree with the reviewer that SENP1 hepatic KO mice develop apoptosis and inflammation first and then steatosis, which is not typical NASH, though SENP1 deficiency does promote NASH pathogenesis in an HFD-induced NAFLD model. Following the reviewer's suggestion, we modified the title as "**SENP1 prevents steatohepatitis by suppressing RIPK1-driven apoptosis and inflammation**" to avoid misunderstanding.

2. The author demonstrated the negative correlation between hepatic SENP1 expression and severity of NAFLD in human specimen. SENP1 expression in mice NAFLD or NASH model should be detected to strengthen the results.

Reply: To address this question, we assessed the levels of SENP1 in hepatocytes from mice with NAFLD, and found that SENP1 is reduced in hepatocytes from NAFLD mice when compared to that of control mice (**Fig. R3a** or revised **Supplementary Fig. 1b**). We also observed reduced SENP1 expression in whole liver extracts of mice fed a high-fat diet (HFD) (**Fig. R1b** or revised **Supplementary Fig. 1a**), which is consistent with the observation of SENP1 expression in human samples of NASH patients.

3. *Senp1^{f/f};Alb-Cre* mice are liver specific, while *Ripk1^{D138N/D138N}* mice are whole body RIPK1 inactivation. To exclude the possibility that inactive RIPK1 in other cell (in addition to hepatocyte) rescued the phenotype of *Senp1^{f/f};Alb-Cre* mice, the author should isolated hepatocyte from *Senp1^{f/f};Alb-Cre; Ripk1^{D138N/D138N}* mice and detected steatosis, inflammatory and apoptosis; or using liver-specific AAV to inhibit hepatic *Ripk1* activity and conducted similar measurement.

Reply: To address this question, we isolated hepatocytes from the livers of 8-m-o *Senp1^{f/f}*, *Senp1^{f/f};Alb-Cre* and

Senp1^{fl/fl}, Alb-Cre, Ripk1^{D138N/D138N} mice, and determined steatosis by measuring the intracellular triglycerides (TG) concentration, and apoptosis by immunoblotting cleaved-caspase-3 (CC3), and inflammatory cytokine production by using quantitative PCR. We found that *Senp1^{fl/fl}, Alb-Cre* hepatocytes derived from livers of 8-m-o mice showed elevated TG concentration (**Fig. R8a** or revised **Supplementary Fig. 1n**), RIPK1 activation as marked by p-S166 RIPK1 (**Fig. R8b** or revised **Supplementary Fig. 1f**), apoptosis as marked by CC3 (**Fig. R8b**), and inflammatory cytokine production (including *Tnf* and *Cxcl1*) (**Fig. R8c** or revised **Supplementary Fig. 1l**) as compared to *Senp1^{fl/fl}* control hepatocytes, all of which were reduced in *Senp1^{fl/fl}, Alb-Cre, Ripk1^{D138N/D138N}* hepatocytes (**Fig. R8a-c**). These results suggest that hepatic deletion of SENP1 leads to cell-autonomous RIPK1 kinase-dependent apoptosis, inflammation, and steatosis.

Fig. R8. (a) The relative intracellular TG content in primary hepatocytes isolated from 8-m-o mice with indicated genotypes (n = 5 mice for each genotype). Each dot represents an individual mouse. Mean \pm s.e.m. One-way ANOVA, post hoc Dunnett's test. (b) Immunoblotting analysis of RIPK1 activation, apoptosis as well as necroptosis in primary hepatocytes isolated from 8-m-o mice with indicated genotypes. (n=4 mice for *Senp1^{fl/fl}*, n=3 mice for *Senp1^{fl/fl}, Alb-Cre* and *Senp1^{fl/fl}, Alb-Cre, Ripk1^{D138N/D138N}*). *: non-specific band. (c) Quantitative RT-PCR analysis of the mRNA expression of *Tnf* and *Cxcl1* from primary hepatocytes isolated from 8-month-old mice of indicated genotypes (n = 5 mice for each genotype). Each dot represents an individual mouse. Mean \pm s.e.m. One-way ANOVA, post hoc Dunnett's test.

4. The author provided solid data to prove that sumoylation of RIPK1 in TNF-RSC promoted its ubiquitylation and activation, and also successfully identified the sumoylated residue in RIPK1. Given sumoylation of RIPK1 in TNF-RSC reached its peak in 5min upon TNF α treatment in WT cells (fig. 5d), the author should explain why SENP1 was recruited at this time course (fig. 5e). Perhaps, PIAS1 was recruited first to activate RIPK1, and then SENP1 involved in to limit signal overamplification? The author should test this sequence by detecting whether PIAS1 was recruited before 5 min, and then SENP1.

Reply: To explore whether PAIS1 is recruited to TNF-RSC before SENP1, we performed a more detailed time course of TNF-RSC, and assessed the levels of PIAS1 and SENP1 in TNF-RSC. We found that both SENP1 and PIAS1 were recruited to TNF-RSC in 2 min, reached their peaks in 5 min, maintained this peak in 15 min, and reduced in 30 min (**Fig. R9**). TNF-RSC is a very transient complex, which assembles in minutes after TNF α stimulation. Usually, TNF-RSC formation reaches its peak in 5 min, when most of the components of TNF-RSC are recruited. For example, RIPK1 ubiquitination is peaked in 5 min in TNF-RSC, while the M1 ubiquitinases HOIP/SHARPIN, the K63 ubiquitinases TRAF2/cIAP1 and the deubiquitinase CYLD also peaked in 5 min in TNF-RSC (**Fig. R10**). We further explored the recruitment of SENP1 in PIAS1-KD cells, and the recruitment of

PIAS1 in SENP1-KD cells to determine the sequence of PIAS1 and SENP1 recruitment. We found that the recruitment of SENP1 in TNF-RSC is reduced in PIAS1-KD cells when compared to PIAS1-WT cells (**Fig. R11a** or revised **Supplementary Fig. 8g**), while the recruitment of PIAS1 is not altered in SENP1-KD cells (**Fig. R11b** or revised **Supplementary Fig. 8h**), suggesting that PIAS1 is at least partially required for the recruitment of SENP1. Because SENP1 is known to interact with SUMO2 noncovalently, which is necessary for its deSUMOylation activity (Ambaye, et al. *Acta Crystallogr F*. PMID:31045562), PIAS1 might be recruited first to perform SUMOylation on RIPK1, then the SUMO chains on RIPK1 increased SENP1 recruitment.

**Fig. R9.** MEFs were stimulated with Flag-TNFα (100 ng/ml) for the indicated times. The recruitment of PIAS1 and SENP1 in TNF-RSC was immunoprecipitated via α-Flag beads and analyzed by western blot.

Draber, et al. *Cell Rep*. PMID:26670046

Feltham, et al. *Cell Rep*. PMID:29642005

**Fig. R10.** Published data suggests that RIPK1 ubiquitination, the M1 ubiquitinases HOIP/SHARPIN, the K63 ubiquitinases TRAF2/cIAP1 and the deubiquitinase CYLD all peaked in 5 min in TNF-RSC after TNFα stimulation.

Fig. R11. PIAS1-WT and PIAS1-KO (a) or SENP1-WT and SENP1-KO (b) MEFs were stimulated with Flag-TNF α (100 ng/ml) for the indicated times. The recruitment of PIAS1 and SENP1 in TNF-RSC was immunoprecipitated via α -Flag beads and analysed by western blots.

5. In according to the current data, PIAS1 was required for RIPK1 activation. However, it is well-established that PIAS1 negatively regulated inflammation. Please make discussion about this difference.

Reply: We have followed the reviewer's suggestion, and made discussion about the difference of PIAS1-regulated inflammation in NF- κ B pathway and RIPK1 kinase-dependent inflammatory pathway as bellow: "Our study defines a novel molecular contribution of PIAS1 that promotes the activity of RIPK1 kinase in mediating apoptosis in SENP1-deficient cells, which is distinct from the role of PIAS1 in limiting inflammation in canonical NF- κ B pathway. Such difference can be observed in other regulators of both NF- κ B pathway and the kinase activity of RIPK1. For example, TAK1 is a positive regulator of NF- κ B-dependent inflammation (Wang, et al. *Nature*, PMID:11460167), and also an endogenous inhibitor of RIPK1 kinase activity (Geng, et al. *Nat Commun*, PMID:28842570). Ablation of TAK1 leads to suppress of canonical NF- κ B-mediated inflammatory response, but promotes RIPK1 kinase-dependent apoptosis due to unleashed RIPK1 activation (Geng, et al. *Nat Commun*, PMID:28842570)."

6. To test the importance of desumoylation activity of SENP1 in the current model, SENP1 and its catalytically inactive C603S mutant (SENP1-C603S) should be replenished in the *Senp1^{fl/fl};Alb-Cre* mice to test RIPK1 activation, and steatosis, inflammatory and apoptosis phenotype in the liver.

Reply: To address this question, we used hepatocyte-specific AAV to reconstitute WT SENP1 and catalytically inactive C599S mutant SENP1 (the catalytic site of human and mouse SENP1 is C603 and C599, respectively) in the hepatocytes of 2-m-o *Senp1^{fl/fl};Alb-Cre* mice. After AAV infection for 2 months (**Fig. R12a** or revised **Supplementary Fig. 2a**), we tested RIPK1 activation, steatosis, inflammatory cytokine production, and apoptosis in the livers of *Senp1^{fl/fl};Alb-Cre,AAV-Senp1^{WT}* (SENP1-WT) and *Senp1^{fl/fl};Alb-Cre,AAV-Senp1^{C599S}* (SENP1-C599S) mice. We detected RIPK1 activation in livers of SENP1-C599S mice, but not that of SENP1-WT mice (**Fig. R12b** or revised **Supplementary Fig. 2b**). Consistently, SENP1-C599S but not SENP1-WT livers showed CC3⁺ cells (**Fig. R12c** or revised **Supplementary Fig. 2c**). SENP1-C599S livers showed elevated inflammatory cytokine production than that of SENP1-WT livers (**Fig. R12d** or revised **Supplementary Fig. 2d**). SENP1-C599S but not SENP1-WT livers display apparent steatosis (**Fig. R12e** or revised **Supplementary Fig. 2e**). Taken together, these results suggest that the deSUMOylation activity of SENP1 is important in suppressing RIPK1 activation, and subsequent apoptosis, inflammation, and steatosis, *in vivo*. We thank the reviewer for this excellent suggestion, which strengthened the paper.

Fig. R12. (a) Fluorescent microscopy images of livers two months after injections with AAV8-TBG-EGFP-P2A-SEN1-WPRE or AAV8-TBG-EGFP-P2A-SEN1(C599S)-WPRE. The construct expresses EGFP and SEN1 linked by self-cleaving P2A peptide. The expression of EGFP allows for verification of the expression of SEN1 and efficacy of infection. (b, c) p-S166 RIPK1 (b) and CC3 (c) staining of liver sections from *Senp1^{fl/fl}; Alb-Cre* control mice, *Senp1^{fl/fl}; Alb-Cre*, AAV-*Senp1^{WT}* (SEN1-WT) and *Senp1^{fl/fl}; Alb-Cre*, AAV-*Senp1^{C599S}* (SEN1-C599S) mice. Graph depicting numbers of p-S166 RIPK1⁺ staining (b) and numbers of CC3⁺ cells (c) on liver sections of indicated genotypes, respectively. Each dot represents an individual mouse. Mean \pm s.e.m. One-way ANOVA, post hoc Dunnett's test. (d) Quantitative RT-PCR analysis of the mRNA expression of cytokines and chemokines in livers of mice from b (n = 5 mice for each genotype). Each dot represents an individual mouse. Mean \pm s.e.m. One-way ANOVA, post hoc Dunnett's test. (e) Oil Red O staining of lipids in liver sections from mice of indicated genotypes. Representative images out of n = 5 mice for each genotype are represented. Graph depicting percentage of lipid content on liver sections of indicated genotypes. Each dot represents an individual mouse. Mean \pm s.e.m. One-way ANOVA, post hoc Dunnett's test.

7. The writing of the manuscript should be improved, with some mistakes need to be revised. such as: line 130, hepatic "SNEP1" deletion. Experimental results should be described in order. For example, figure 1e was described after Figure 1f and 1g by the author.

Reply: We have corrected this typo in line 130, and changed the order of Fig. 1e-g.

8. Sample size should be expanded, especially for the in vivo measurements in fig. 3 and fig. 7, in which only three mice were included in some groups.

Reply: We have expanded the sample size and repeated these experiments (revised **Fig. 3** and **Fig. 7**). The results are consistent to our original results.

Reviewer #3 (Remarks to the Author):

In this manuscript Yan et al decipher the role and the regulation of RIPK1-SUMOylation which was hitherto unknown and show the protective role of SENP1 in hepatocyte during liver inflammation in NASH. They performed several in vitro and in vivo studies to understand the regulation of RIPK1 SUMOylation by SENP1 and PIAS1 and its incidence on TNF-induced cell death. They found that de-SUMOylation of RIPK1 by SENP1 inhibits its activation and TNF α -induced apoptosis. Conversely, in SENP1 KO PIAS1 promotes RIPK1 SUMOylation and sensitizes to TNF induced cell death. Overall, data are convincing and offer a persuasive mechanism on the unknown role of RIPK1 SUMOylation.

Reply: We thank this reviewer for strong support and suggestions that helped us to strengthen aspects of this manuscript! We have carefully considered and addressed every comment from this reviewer.

However, some conclusions are overstated and additional experiments are necessary to improve the manuscript. The authors demonstrate the protective role of SENP1 in NASH. Although the reduction of SENP1 during disease progression is convincing, it is unlikely that the role of SENP1 is restricted to metabolic liver disease. SENP1 inhibits RIPK1 apoptosis induced by TNF α stimulation. As TNF α is a very common cytokine in the inflamed liver (no matter the etiology), it would be expected that SENP1 would protect hepatocyte from cell death in other type of liver disease. To confirm this hypothesis, it would be interesting to see if LPS triggers liver injury in mice as well. Statements in the manuscript should be modified accordingly.

Reply: To address this question, we explored the role of SENP1-RIPK1 axis in LPS-induced acute liver injury (ALI). We used 2-m-o mice, when spontaneous liver damage is not observed in *Senp1^{fl/fl}, Alb-Cre* mice (**Fig. R13a** or revised **Supplementary Fig. 9a**). After LPS injection for 4 hours, we observed substantial liver damage in *Senp1^{fl/fl}, Alb-Cre* mice compared to *Senp1^{fl/fl}* control mice as determined by ALT, AST levels and histology stains (**Fig. R13b, c** or revised **Supplementary Fig. 9b, c**), TUNEL⁺ cells (**Fig. R13d** or revised **Supplementary Fig. 9d**), and p-RIPK1⁺ cells (**Fig. R13d**). In addition, the mRNA levels of inflammatory cytokines, including *Tnf*, *Il1a*, *Il1b*, *Il6*, *Ccl2* and *Ccl5* were substantially higher in *Senp1^{fl/fl}, Alb-Cre* livers than that in *Senp1^{fl/fl}* livers (**Fig. R13e** or revised **Supplementary Fig. 9e**). Strikingly, RIPK1-D138N mutation largely reduced the liver damage and inflammation in *Senp1^{fl/fl}, Alb-Cre* mice (**Fig. R13b-e**). Taken together, these results demonstrated the essential role of RIPK1 kinase activity in mediating liver inflammation and damage in LPS-induced ALI, under a hepatocyte-specific SENP1-deficiency condition.

Fig. R13. (a) Serum levels of ALT and AST of 2 months old mice with indicated genotypes ($n = 5$ mice for each genotype). Each dot represents an individual mouse. Mean \pm s.e.m. One-way ANOVA, post hoc Dunnett's test. (b) Plasma samples were harvested at 4 h after LPS (50 ng/g) injection for the measurement ALT and AST in 2 months old mice with indicated genotypes ($n = 5$ for each genotype). Each dot represents an individual mouse. Mean \pm s.e.m. One-way ANOVA, post hoc Dunnett's test. (c) HE staining of liver sections of mice from b. Representative images out of $n = 5$ mice for each genotype are represented. (d) TUNEL assay and p-S166 RIPK1 staining of liver sections from mice treated with LPS. Graph depicting numbers of TUNEL⁺ cells and percentage of p-S166 RIPK1⁺ staining on liver sections of indicated genotypes, respectively. Each dot represents an individual mouse. Mean \pm s.e.m. One-way ANOVA, post hoc Dunnett's test. (e) Quantitative RT-PCR analysis of the mRNA expression of cytokines and chemokines in livers of mice from b ($n = 5$ mice for each genotype). Each dot represents an individual mouse. Mean \pm s.e.m. One-way ANOVA, post hoc Dunnett's test.

The authors claim that loss of SENP1 promote lipid deposition, however the oil red staining (only marker used in this study) does not seem to show specific staining for lipid droplets but conversely the all tissue seems to be red. Addition experiments such as liver triglyceride and liver cholesterol measurements are needed for this statement.

Reply: To address this question, we performed additional experiments as suggested by the reviewer to measure the lipid content in these mice. Consistently, 8-m-o *Senp1^{fl/fl}; Alb-Cre* mice showed higher hepatic concentrations of triglyceride (TG) and total cholesterol (TC) than that of age-matched *Senp1^{fl/fl}* mice, which were reduced in *Senp1^{fl/fl}; Alb-Cre; Ripk1^{D138N/D138N}* mice (Fig. R14a or revised Fig. 2e). In addition, in HFD-induced NAFLD model, *Senp1^{fl/fl}; Alb-Cre* mice also showed increased hepatic concentrations of TG and TC as compared with that

of *Senp1^{fl/fl}* mice, which were reduced in livers of *Senp1^{fl/fl};Alb-Cre;Ripk1^{D138N/D138N}* mice (**Fig. R14b** or revised **Supplementary Fig. 9i**). We thank the reviewer for this suggestion, which strengthened the conclusions.

Fig. R14. Liver lipid content, including TG, TC content per gram of liver from mice with indicated genotypes fed with (a) or without HFD (b) ($n = 5$ mice for each genotype). Each dot represents an individual mouse. Mean \pm s.e.m. One-way ANOVA, post hoc Dunnett's test.

All the *in vitro* studies were done on Mouse Embryonic Fibroblasts whereas the *in vivo* analysis focus on hepatocytes. In order to link both sections, it would be interesting to see that SENP1 KO primary hepatocytes are equally sensible to TNF α *in vitro* and protected by RIPK1 kinase inhibition.

Reply: To address this question, we stimulated cultured primary hepatocytes derived from 2-m-o *Senp1^{fl/fl}* (SENP1-WT) and *Senp1^{fl/fl};Alb-Cre* (SENP1-KO) mice with TNF α for 24 hours and assessed cell death, RIPK1 phosphorylation levels, and apoptosis. We found that deletion of SENP1 in hepatocytes increased cell death (**Fig. R15a** or revised **Fig. 3d**), RIPK1 activation as determined by its S166 phosphorylation, and apoptosis as shown by increased levels of the cleaved-caspase-3 (CC3) (**Fig. R15b** or revised **Fig. 3e**). Consistently, the death of SENP1-KO hepatocytes, RIPK1 activation and apoptosis induced by TNF α was fully protected by RIPK1-D138N mutation (**Fig. R15a, b**). These results suggest that SENP1 deficiency in hepatocytes promotes RIPK1 activation and apoptosis in response to TNF α *in vitro*.

Fig. R15. (a, b) SENP1 deficiency sensitized primary hepatocytes to cell death induced by TNF α . Primary hepatocytes derived from 2-m-o *Senp1^{fl/fl}*, *Senp1^{fl/fl};Alb-Cre*, *Senp1^{fl/fl};Alb-Cre;Ripk1^{D138N/D138N}* mice were treated with 20 ng/ml TNF α for indicated time. Cell death was measured as a function of time by SytoxGreen positivity assay, data are represented as mean \pm s.d. of $n = 3$ independent samples of one representative experiment out of three independent experiments (a). The levels of p-S166 RIPK1 and

CC3 were determined by immunoblotting (b).

Extended 2E, the authors claim that SENP1 does not affect NF- κ B activation. However, knockdown with shSenp1-2 (the most potent shRNA regarding SENP1 expression) seems to reduce p-I κ B α (S32) and p-JNK. These results should be confirmed and quantified.

Reply: We have performed the experiment with additional repeats and provided quantitation data on p-I κ B α and p-JNK (Fig. R16 or revised Supplementary Fig. 3e). We observed slight decrease of p-JNK at 30 min after TNF α treatment in shSenp1-2 cells. We modified the claim as “SENP1 deficiency has minor effect on NF- κ B and MAPK pathway activation”.

Fig. R16. MEFs stably expressing the indicated shRNA were stimulated with 10 ng/ml TNF α for indicated periods of time and the whole-cell lysates were immunoblotted as indicated. Quantitative data are represented as mean \pm s.e. m. of three independent experiments.

The authors show that SENP1 KO cells are sensitized to TNF-induced apoptosis or necroptosis, when caspases are inhibited. As apoptosis and necroptosis are two cell death mediators observed in NASH, what type of cell death is induced in Senp1 $^{f/f}$;Alb-Cre mice?

Reply: To address this question, we isolated hepatocytes from the livers of 8-m-o WT (*Senp1 $^{f/f}$*), *Senp1 $^{f/f}$* ,*Alb-Cre* and *Senp1 $^{f/f}$* ,*Alb-Cre*,*Ripk1 $^{D138N/D138N}$* mice, and determined apoptosis by immunoblotting cleaved-caspase-3 (CC3), and necroptosis by immunoblotting p-RIPK3(T231/S232), the hallmark of necroptosis. We were able to detect increased p-RIPK1 and CC3 in *Senp1 $^{f/f}$* ,*Alb-Cre* hepatocytes, which was blocked in *Senp1 $^{f/f}$* ,*Alb-Cre*,*Ripk1 $^{D138N/D138N}$* hepatocytes (Fig. R8b or revised Supplementary Fig. 1f). However, we did not detect apparent p-RIPK3(T231/S232) signals in *Senp1 $^{f/f}$* ,*Alb-Cre* hepatocytes (Fig. R8b), suggesting apoptosis is induced by RIPK1 activation in *Senp1 $^{f/f}$* ,*Alb-Cre* mice.

Minor comments:

-Line 130: change SNEP1 to SENP1

Reply: We have corrected this typo.

-Method for preparing frozen blocks is missing

Reply: We have included the method for preparing frozen blocks (in “Oil red O staining”).

-Statistical analysis: Does the distribution of the samples was checked before statistical analysis? Test should be different if the samples are normally distributed or not.

Reply: Normality of the samples were all checked by using the Shapiro-Wilk test before statistical analysis. For normal distribution, either unpaired two-tailed Student’s t-test for comparison between two groups, or one-way ANOVA with post hoc Dunnett’s tests for comparisons among multiple groups with a single control was applied, while for non-normal distribution, non-parametric test (Mann-Whitney test) was used instead. This has also been described in Method section in revised manuscript.

- Figure 1e – DAPI can be hardly seen – provide a higher exposure picture. In addition, RIPK1 (S166) staining seem to stain non-parenchymal cells. Did p-RIPK1 was also observed in hepatocytes?

Reply: We have provided a higher exposure picture of Fig. 1e (**Fig. R17** or revised **Fig. 1g**). Since RIPK1 activation in hepatocyte can drive apoptosis in the absence of SENP1, which may lead to cell shrinkage, or the loss of cell volume, the p-RIPK1⁺ cells may appear smaller than normal hepatocyte. We also detected p-RIPK1 by western blotting to confirm the activation of RIPK1 in hepatocyte. We isolated hepatocytes from the livers of 8-m-o WT (*Senp1^{fl/fl}*), *Senp1^{fl/fl}, Alb-Cre* and *Senp1^{fl/fl}, Alb-Cre, Ripk1^{D138N/D138N}* mice, and detected p-RIPK1(S166). We observed increased p-RIPK1 in *Senp1^{fl/fl}, Alb-Cre* hepatocytes, which was blocked in *Senp1^{fl/fl}, Alb-Cre, Ripk1^{D138N/D138N}* hepatocytes (**Fig. R8b** or revised **Supplementary Fig. 1f**)

Fig. R17. Immunofluorescence images of p-S166 RIPK1 of liver sections from 8-month-old mice with indicated genotypes.

REVIEWERS' COMMENTS

Reviewer #1 (Remarks to the Author):

The authors have made a huge effort to address all the comments and remarks resulting in an overall improved manuscript that as far as this Reviewer concern is suitable for publications.

Reviewer #2 (Remarks to the Author):

The author has excellently addressed all my comments. I have no questions further.

Reviewer #3 (Remarks to the Author):

The authors provided substantial work to answer the reviewer questions and strengthen their results and the manuscript.

Comments:

Line 160 to 174. The authors should rewrite this section of their manuscript. Desmin staining cannot be done to "explore how RIPK1 kinase is involved in promoting Senp1f/f Alb-Cre-induced liver fibrosis". Desmin staining only shows an increase of HSC in the liver. However, Masson's trichrome staining shows an increase of liver fibrosis consistent with the upregulation of the HSC activation genes Col1a1, and Col3a1.

Figure R8, the authors investigated if SENP1 deletion can promotes necroptosis by looking at RIPK3 phosphorylation. The figure is missing in the Supplementary Figure 1. In addition, in their WB, they should have added a positive control to confirm the absence of pRIPK3 and as it is difficult to demonstrate negative results look at pMLKL as they did in Supplementary Fig4.

Point-to-Point Responses to the Reviewer 3's Comments (#NCOMMS-22-10617-A)

Reviewer #3 (Remarks to the Author):

The authors provided substantial work to answer the reviewer questions and strengthen their results and the manuscript.

Comments:

1. Line 160 to 174. The authors should rewrite this section of their manuscript. Desmin staining cannot be done to “explore how RIPK1 kinase is involved in promoting *Senp1^{f/f}* Alb-Cre-induced liver fibrosis”. Desmin staining only shows an increase of HSC in the liver. However, Masson's trichrome staining shows an increase of liver fibrosis consistent with the upregulation of the HSC activation genes *Coll1a1*, and *Col3a1*.

Reply: Hepatocytic damage and liver inflammation are known to activate hepatic stellate cells (HSC), which are the major source of fibrosis in the liver (PMID: 15690074, 18471545, 28045404). We found that inhibition of RIPK1 by *Ripk1^{D138N/D138N}* knockin mutation suppressed hepatocytic damage and liver inflammation in *Senp1^{f/f};Alb-Cre* mice, thus fibrosis and the activation of HSC that were increased in *Senp1^{f/f};Alb-Cre* livers were also reduced in *Senp1^{f/f};Alb-Cre;Ripk1^{D138N/D138N}* livers probably due to reduced hepatocytic damage and liver inflammation. We have revised this section and clarified this claim. We thank the reviewer for this suggestion!

2. Figure R8, the authors investigated if SENP1 deletion can promotes necroptosis by looking at RIPK3 phosphorylation. The figure is missing in the Supplementary Figure 1. In addition, in their WB, they should have added a positive control to confirm the absence of pRIPK3 and as it is difficult to demonstrate negative results look at pMLKL as they did in Supplementary Fig4.

Reply: We thank the reviewer for this suggestion! We have included a positive control of p-RIPK3 in this blot. As shown in figure below (or **Supplementary Fig. 1f** in revised manuscript), we only detected non-specific bands from lysate of all hepatocytes with indicated genotypes in p-RIPK3 blot, further suggesting the absence of necroptosis in SENP1-KO hepatocytes. This is consistent to a recently published paper that RIPK3 is epigenetically silenced in hepatocytes and renders them unable to undergo necroptosis in NASH (Preston et al., *Gastroenterology*, 2022. PMID: 36037995).

Fig. R1, Immunoblotting analysis of RIPK1 activation, apoptosis as well as necroptosis in primary hepatocytes isolated from 8-m-o mice with indicated genotypes. (n=4 mice for *Senp1^{f/f}*, n=3 mice for *Senp1^{f/f};Alb-Cre* and *Senp1^{f/f};Alb-Cre;Ripk1^{D138N/D138N}*). Lysate from MEFs treated with T(10 ng/ml)/CHX(C, 2 µg/ml)/zVAD(Z, 10 µM) for 4 hr were used as a positive control for p-RIPK3(T231/S232) signal. *: non-specific band.